# Life-cycle-related gene expression patterns in the brown algae

Pélagie Ratchinski[1], Olivier Godfroy[2], Benjamin Noel[3], Jean-Marc Aury[3], J Mark Cock[1]*

[1]Sorbonne Université, CNRS, Algal Genetics Group, Integrative Biology of Marine Models Laboratory, Roscoff, France; [2]Sorbonne Université, CNRS, CMAR, Integrative Biology of Marine Models Laboratory, Roscoff, France; [3]Génomique Métabolique, Genoscope, Institut François Jacob, CEA, CNRS, Université Evry, Université Paris-Saclay, Evry, France

**eLife Assessment**

This manuscript presents an in-depth analysis of gene expression across multiple brown algal species with differing life histories, providing **convincing** evidence for the conservation of life cycle-specific gene expression. While largely descriptive, the study is an **important** step forward in understanding the core cellular processes that differ between life cycle phases, and its findings will be of broad interest to developmental and evolutionary biologists.

*For correspondence:
cock@sb-roscoff.fr

Competing interest: The authors declare that no competing interests exist.

**Abstract** Brown algae are important primary constituents of marine coastal ecosystems, characterised by complex life cycles and various levels of complex multicellular development. However, the molecular processes that underlie development and life cycle progression in the brown algae remain poorly understood. In this study, pairwise comparisons of gametophyte and sporophyte transcriptomes across 10 diverse brown algal species showed that the total number of genes exhibiting generation-biased or generation-specific expression in each species was correlated with the degree of dimorphism between life cycle generations. However, analysis of gene ontology terms assigned to the generation-biased/generation-specific genes indicated that each generation (i.e. the sporophyte and the gametophyte) also has characteristic broad life-cycle-related features that have been conserved during evolution. A more detailed analysis of *Ectocarpus* species 7 identified progressive transcriptome changes over its entire life cycle, with a particularly marked change in transcriptome composition during the first day of sporophyte development, characterised by downregulation of flagellar and transcription factor genes and upregulation of a subset of translation genes. Comparison with a similar transcriptomic time series for the evolutionarily distant (about 250 My) brown alga *Dictyota dichotoma* indicated considerable conservation of co-expressed gene modules between the two species, particularly for modules that were enriched in genes assigned to evolutionarily conserved functional categories. This study therefore identified broad life-cycle- and development-related patterns of gene expression that are conserved across the brown algae.

## Introduction

Brown algae are major primary components of diverse and widespread coastal ecosystems, often forming extensive underwater forests (*Bringloe et al., 2020*; *Eger et al., 2023*). The group includes large (up to 50 m) multicellular organisms that rival many land plant species in complexity. From a developmental point of view, brown algae are particularly interesting because they evolved complex multicellularity independently of land plants and animals (*Cock et al., 2010*). Taxonomically, these

**eLife digest** Coastal marine environments are often populated by large underwater forests of brown algae, such as seaweeds and kelps. The brown algae are key components of these coastal ecosystems, providing food and habitat for numerous organisms.

From a developmental perspective, brown algae are particularly interesting because they evolved multicellularity independently of land plants and animals. They therefore represent an alternative system to investigate the evolution and molecular bases of developmental processes.

Most brown algae have life cycles that alternate between a spore-producing generation (the sporophyte) and a gamete-producing generation (the gametophyte). In some cases, these distinct life stages may enhance survival under variable conditions. However, the genetic basis of development in these complex multicellular organisms – which could shed more light on their ecology – remains poorly understood.

Ratchinski et al. investigated the activity of genes across multiple life-cycle stages in a broad range of brown algal species to identify groups of genes that potentially play key roles at specific stages of development.

The results showed that brown algal species with strongly differentiated life-cycle generations tended to have more genes that are differentially expressed during each generation, correlating with the differences in morphological traits.

Furthermore, several groups of genes that exhibited coordinated patterns of expression over the course of the life cycle (so-called co-expression modules) tended to share related functions. Ratchinski et al. also noted extensive changes in gene expression during the early stages of sporophyte development, and similar patterns were observed in two evolutionarily distant brown algae species.

Seaweed cultivation is attracting growing interest for producing biomass for human consumption and other applications. At the same time, wild populations of brown algae are increasingly threatened by global warming. A deeper understanding of brown algae biology is therefore vital for developing sustainable cultivation methods and effective conservation strategies to protect natural brown algal ecosystems.

seaweeds are grouped into the class Phaeophyceae, which is a major taxon within the stramenopiles, and therefore very distant, phylogenetically, from the Archaeplastida and the Opisthokonta lineages that gave rise to land plants and animals, respectively. Consequently, brown algae represent interesting alternative systems to investigate the evolution and molecular bases of developmental processes.

Most brown algae have haploid-diploid life cycles involving an alternation between two independent multicellular generations, the sporophyte and the gametophyte (*Cock et al., 2014*). These seaweeds are therefore capable of deploying two different multicellular developmental programs, each at the appropriate stage of their complex life cycles. The ecological functions of brown algal haploid-diploid life cycles can be difficult to study, particularly when one of the generations is microscopic. However, there is evidence, in kelps for example, that microscopic stages (gametophytes, but possibly also very young sporophyte stages) may have a survival function, allowing a population to persist when the macroscopic sporophyte stage is not present (*Carney and Edwards, 2006*). A study of populations of filamentous brown algae of the genus *Ectocarpus* in the field indicated that gametophytes grew epiphytically on the brown alga *Scytosiphon promiscuus* (then classified taxonomically as *Scytosiphon lomentaria*) in the spring, whereas sporophytes grew throughout the year on abiotic substrata. These observations indicate that the sporophyte may be the most important generation for persistence of the population in this genus (*Couceiro et al., 2015*). Note, however, that a principally asexual population of *Ectocarpus siliculosus* located at a second site inhabited both epiphytic and epilithic niches, indicating that the relationship between life cycle generation and niche may be complex (*Couceiro et al., 2015*). The phenology of the brown alga *D. dichotoma* also appears to be complex with, for example, overlapping generations and sporophytes and gametophytes reported to occur simultaneously (*Tronholm et al., 2008*).

The relative sizes of the two life cycle generations vary considerably across brown algae, with either the sporophyte or the gametophyte being the largest generation (dimorphism) or the two generations

having very similar sizes and morphologies (isomorphy). Transitions between life cycles with different degrees of dimorphism have occurred frequently during the evolution of the brown algae (*Cock et al., 2014*). Variations on this central sexual life cycle have been observed in culture for several species, and these variations can provide insights into the mechanisms that control life cycle progression. For example, if *Ectocarpus* gametes fail to fuse with a gamete of the opposite sex to form a zygote, they are often capable of germinating parthenogenetically to produce a haploid partheno-sporophyte that is morphologically and functionally indistinguishable from a diploid sporophyte (*Müller, 1967*). The existence of these haploid sporophytes indicates that life cycle generation identity is not determined by ploidy in this species and is therefore presumably under genetic control.

The occurrence of two multicellular generations during the life cycles of most brown algae allows genetic screens to be carried out for mutations that cause switching between the development programs of the two generations, that is mutations that lead to deployment of the incorrect program at a specific point in the life cycle (*Coelho and Cock, 2020*). Using this approach, two three-amino-acid-loop-extension (TALE) homeodomain transcription factor (HD TF) genes, *OUROBOROS (ORO)* or *SAMSARA (SAM)*, have been shown to be necessary for deployment of the sporophyte genera-tion developmental program in *Ectocarpus* (*Coelho et al., 2011*; *Arun et al., 2019*). Mutations in either of these genes suppress the ability of the alga to produce a functional sporophyte, leading to deployment of the gametophyte program at life cycle stages where a sporophyte should normally be produced. In addition, *Ectocarpus* sporophytes have been shown to produce a diffusible factor that can induce sporophyte development in gametophyte initial cells (*Arun et al., 2013*; *Yao et al., 2021*). The diffusible factor does not induce life cycle generation switching in *oro* or *sam* mutant gameto-phyte initial cells, indicating that it acts upstream of ORO and SAM (*Arun et al., 2013*; *Arun et al., 2019*).

The genetic pathways that underlie the deployment of the sporophyte and gametophyte devel-opmental programs in brown algae are still poorly understood. Mutations in a small number of genes have been shown to lead to developmental defects in *Ectocarpus* (*Macaisne et al., 2017*; *Godfroy et al., 2017*; *Godfroy et al., 2023*) but a mutation in the *IMMEDIATE UPRIGHT* gene is the only lesion that specifically affects the development of one of the two generations (*Peters et al., 2008*). Transcriptomics has been used to identify genes that are differentially expressed between the two generations in *Ectocarpus* (*Arun et al., 2019*; *Lipinska et al., 2019*; *Bourdareau et al., 2021*). These transcriptomic analyses focused on comparisons of adult sporophyte and gametophyte thalli and have allowed the identification of several hundred genes that are differentially expressed between these two stages of development. However, the analyses were not designed to detect genes that are important during early development and do not provide any information about the kinetics of gene expression over development. In recent years, several detailed studies have generated gene expression data for multiple developmental stages or organs for several different brown algal species (*Pearson et al., 2019*; *Shao et al., 2019*; *Shan et al., 2020*; *Zhang et al., 2021*; *Graf et al., 2022*; *Liang et al., 2023*; *Uji et al., 2024*), including a transcriptomic analysis of early stages of *Dictyota dichotoma* sporophyte development (*Bogaert et al., 2017*). This latter study detected changes in the transcriptome less than 1 hr after fertilisation, providing evidence for de novo transcription in the zygote, and identified a set of genes with diverse functions that were differentially regulated during early sporophyte development.

In the current study, transcriptomic data, recently generated by the Phaeoexplorer project (*Denoeud et al., 2024*), was used to analyse differential gene expression between the adult sporo-phyte and gametophyte generations of a broad range of brown algae. The proportion of genes that showed generation-biased or generation-specific expression across these two stages was correlated with the level of dimorphism between the two generations indicating that the differentially expressed genes are primarily linked to morphological differentiation. However, analysis of gene ontology terms also identified underlying, life-cycle-related characteristics that have been conserved during evolu-tion, with general classes of gene function being conserved across sporophytes of the 10 species, and different general classes of gene function being conserved across the gametophytes. In addition, analysis of transcriptomic data corresponding to multiple stages of the *Ectocarpus* life cycle, gener-ated by combining published RNA sequencing (RNA-seq) data with newly-generated timepoints, identified co-expression modules containing genes with specific expression patterns during life cycle progression. This analysis identified several modules that exhibited marked expression pattern

changes during early sporophyte evolution, providing insights into key genetic events during this period. Comparison with a set of gene co-expression modules generated for *D. dichotoma* identified conserved features shared by these two distantly related brown algal species.

## Results

### Across the brown algal lineage, the proportion of generation-biased genes is correlated with the degree of life cycle dimorphism

RNA-seq data for the adult sporophyte and gametophyte generations of 10 different brown algal species (*Figure 1A*, *Supplementary file 1*) were analysed to obtain a broad overview of generation-biased gene expression across the brown algae. The species analysed had markedly different haploid-diploid life cycles, ranging from sporophyte dominance, through sporophyte-gametophyte isomorphy to gametophyte dominance (*Figure 1—figure supplement 1*). As previously observed in a smaller-scale analysis that looked at four different brown algal species (*Lipinska et al., 2019*), the proportion of generation-biased/generation-specific genes (see Materials and methods for definitions of generation-biased and generation-specific genes) identified in each species was positively correlated with the degree of intergenerational dimorphism (*Figure 1B*, *Figure 1—figure supplement 1*, *Supplementary file 2*). Surprisingly, however, species with strongly dimorphic life cycles had greater numbers of both sporophyte- and gametophyte-biased/specific genes, indicating either that gene downregulation plays an important role in the establishment of dimorphic generations or that there is also cryptic complexification of the more morphologically simple generation.

We also noted that, depending on the species, only between 10% and 24% (*Supplementary file 2*) of genes were expressed during one generation but not the other (defined for this analysis as mean TPM $\geqslant$1 for one generation but mean TPM <1 in the other generation), indicating that most genes are expressed during both generations of the life cycle. There is therefore considerable scope for many genes to have functions during both generations of the life cycle. In terms of selection, this observation indicates that most genes are potentially directly exposed to purifying selection during the haploid phase (where there is no effect of masking by a second allele of each locus) and few genes are expressed solely in a diploid context where masking would be most effective.

### The sporophyte and gametophyte generations express characteristic general functional categories of genes

To evaluate the degree of conservation of generation-biased/specific gene sets across the 10 brown algal species, an orthogroup analysis was carried out (*Supplementary file 3*) using Orthofinder (*Emms and Kelly, 2019*). Pairwise comparisons of generation-biased/specific genes that had orthologues in all 10 species indicated that the sets of generation-biased/specific genes were poorly conserved across species, even for closely-related species such as the two *Saccharina* spp. (*Figure 1C*). None of the orthogroups exhibited conserved generation bias across all the 10 species. The lack of conservation of generation-biased/specific gene sets across species is in line with observations made in the earlier study that compared four brown algal species (*Lipinska et al., 2019*). However, despite the poor conservation of generation-biased/specific expression at the individual gene level, analysis of enriched gene ontology terms across the entire generation-biased/specific gene sets from the 10 studied brown algae identified distinct sets of enriched terms for both the sporophyte-biased/specific and the gametophyte-biased/specific genes (*Figure 2*) and the general classes of enriched gene ontology term were conserved for each individual generation across the 10 species. For example, the sporophyte-biased/specific gene sets tended to be enriched in genes with predicted functions related to membrane transport and metabolism, whereas the gametophyte-biased/specific gene sets tended to be enriched in genes with predicted functions related to flagella biosynthesis and function, protein biosynthesis, and DNA-related functions (*Figure 2*). These analyses identified an underlying similarity between sporophyte generations across species and between gametophyte generations across species, and indicated that these generation-characteristic patterns of gene expression are present even in species where the two generations are isomorphic or nearly isomorphic.

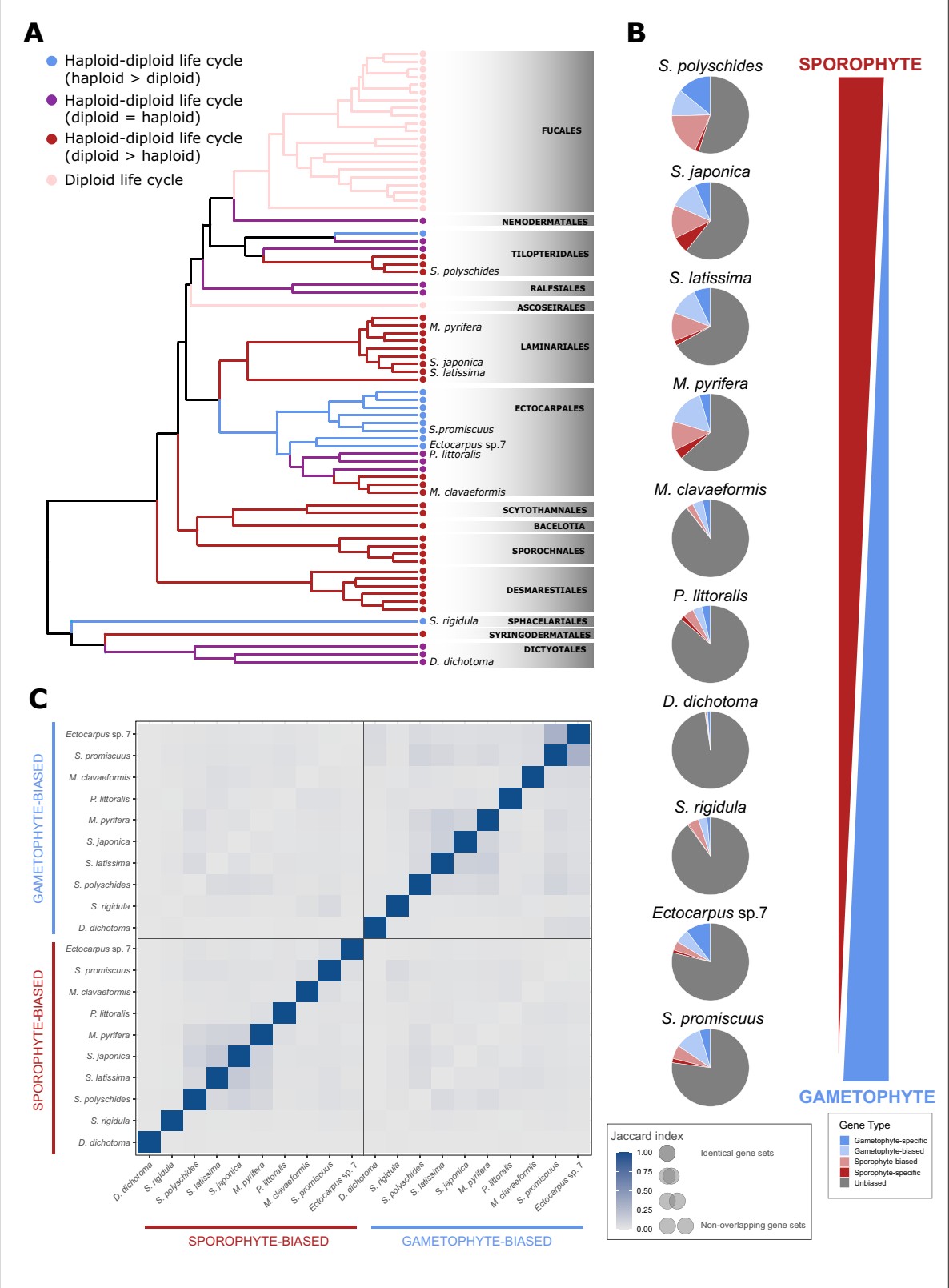

**Figure 1.** Generation-biased gene expression across the brown algae. (**A**) Cladogram of the brown algae showing the 10 species for which generation-biased gene expression was compared. Adapted from ***Cock et al., 2014***. (**B**) Generation-biased gene expression in relation to life cycle dimorphism. Pie diagrams indicating the proportions of sporophyte-specific (dark red), sporophyte-biased (light red), gametophyte-specific (dark blue), gametophyte-biased (light blue), and unbiased (grey) genes in species with different haploid-diploid life cycles ranging from sporophyte-dominant

*Figure 1 continued on next page*

*Figure 1 continued*

through isomorphic to gametophyte-dominant (indicated by the red and blue wedges). sp., species. (**C**) Overlaps (Jaccard index) between sets of gametophyte-biased/specific and sporophyte-biased/specific genes across the 10 species.

The online version of this article includes the following figure supplement(s) for figure 1:

**Figure supplement 1.** Schematic life cycles and photographs of the sporophyte and gametophyte generations of brown algal species analysed in this study illustrating the different degrees of generational dimorphism.

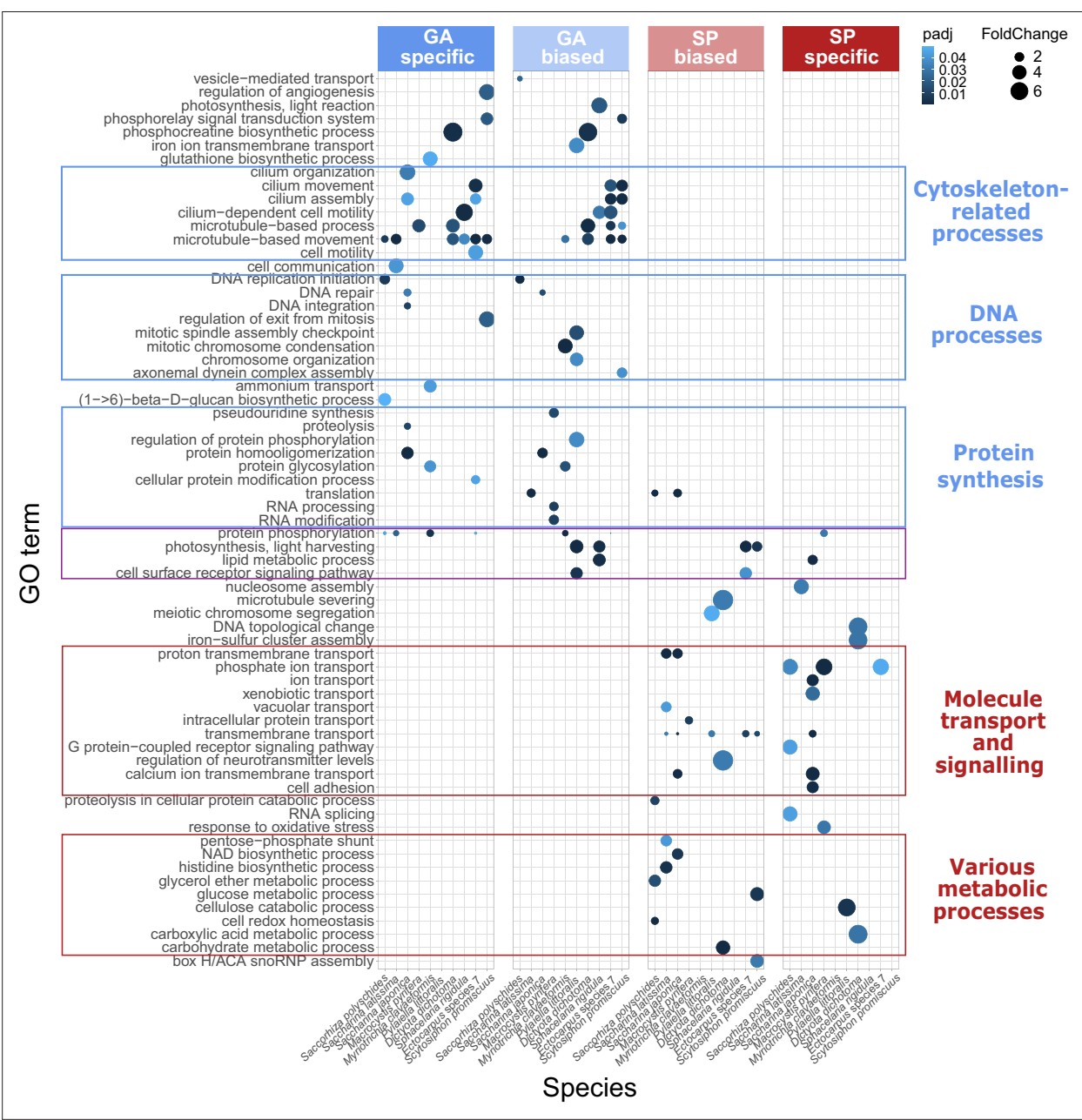

**Figure 2.** Biological function gene ontology term enrichment in the generation-biased/specific gene sets of the 10 species. Conserved general functional categories are indicated by coloured boxes. padj, p-value adjusted for multiple testing based on the Benjamini-Hochberg false discovery rate; GA, gametophyte; SP, sporophyte.

## Changes in gene expression patterns over the course of the *Ectocarpus* species 7 life cycle reflect the cyclic transitions between life cycle stages

The above analyses specifically provided information about gene regulation patterns underlying morphological and functional differences between the adult stages of the two generations. To obtain a broader view of gene expression during the life cycle, RNA-seq samples corresponding to multiple stages of the *Ectocarpus* species 7 life cycle were analysed: replicate samples corresponding to two very early stages (24 hr and 48 hr after gamete release; *Figure 3—figure supplement 2*) of sporophyte development were generated and combined with publicly available samples corresponding to eight other stages of the life cycle (*Supplementary file 1*).

In a Principal Component Analysis (PCA), the dispersion of the samples corresponding to these 10 life cycle stages mirrored the cyclic organisation of the alga life cycle (*Figure 3A*). Together, the two main axes explained 57% of the observed variability, with adult stages grouping together at one end of the first axis and early stages localised at the other end. The second axis approximately separated the sporophyte and gametophyte phases of the life cycle. The PCA pattern indicated gradual changes in transcriptome content between each stage of the life cycle, so that successive stages possessed overlapping but different transcriptomes.

## The first day of sporophyte development corresponds to a major shift in the pattern of gene expression

The PCA also indicated marked differences between the gamete stage and all the other stages of the life cycle (*Figure 3A*). To further investigate this observation, the Trendy R package (*Bacher et al., 2018*) was used to calculate the fit of segmented regression models to individual gene expression patterns in order to identify individual genes that best matched the overall pattern of regression breakpoints. When this approach was applied to the sporophyte developmental series, a good regression fit ($R^2>0.5$) could be calculated for 5196 genes and 32% of these genes exhibited a change in expression (i.e. a regression breakpoint) during the transition from the gamete stage to the sporophyte initial cell (24 h) stage (*Figure 3B*), indicating that this transition involves a major shift in the pattern of gene expression.

To complement the above analysis, DESeq2 (*Love et al., 2014*) was used to identify genes that were differentially expressed between the free-swimming gamete and the sporophyte initial cell stage. This analysis detected 10,955 differentially expressed genes, corresponding to 4372 and 6583 significantly up- and down-regulated genes, respectively. These differentially expressed genes represent 58% of the total number of genes in the *Ectocarpus* species 7 genome (*Figure 3C*). For comparison, when adult sporophyte and gametophyte stages were compared (*Figure 1B*), 3726 upregulated and 2577 downregulated genes (35% of the total number of genes overall) were detected. The DESeq2 analysis therefore supported the conclusion that a marked change in gene expression pattern occurred between the free-swimming gamete stage and the sporophyte initial cell stage.

Interestingly, in the comparison between gametes and the sporophyte initial cell, the mean fold change in transcript abundance for downregulated genes at the transition between free-swimming gamete and initial cell was significantly higher than the mean fold change for genes that were upregulated at this transition (Wilcoxon rank sum test p-values$<2.10^{-16}$; *Figure 3D*). This observation suggested narrower expression ranges for the downregulated genes, and this hypothesis was supported by a comparison of tau values (a measure of breadth of expression) for the downregulated and upregulated genes, which showed that the downregulated genes had significantly higher tau values (Wilcoxon rank sum test p-values$<2.10^{-16}$; *Figure 3—figure supplement 2*). A comparison across the 10 life cycle stages also showed that 61% of the genes downregulated at the gamete to initial cell transition were most strongly expressed at the gamete stage (data in *Supplementary file 4* and *Supplementary file 5*).

## Analysis of gene co-expression modules indicated coordinated regulation of genes associated with several cellular processes

To further analyse gene expression patterns over the life cycle, a weighted correlation network analysis (WGCNA) approach (*Langfelder and Horvath, 2008*) was used to group genes according to

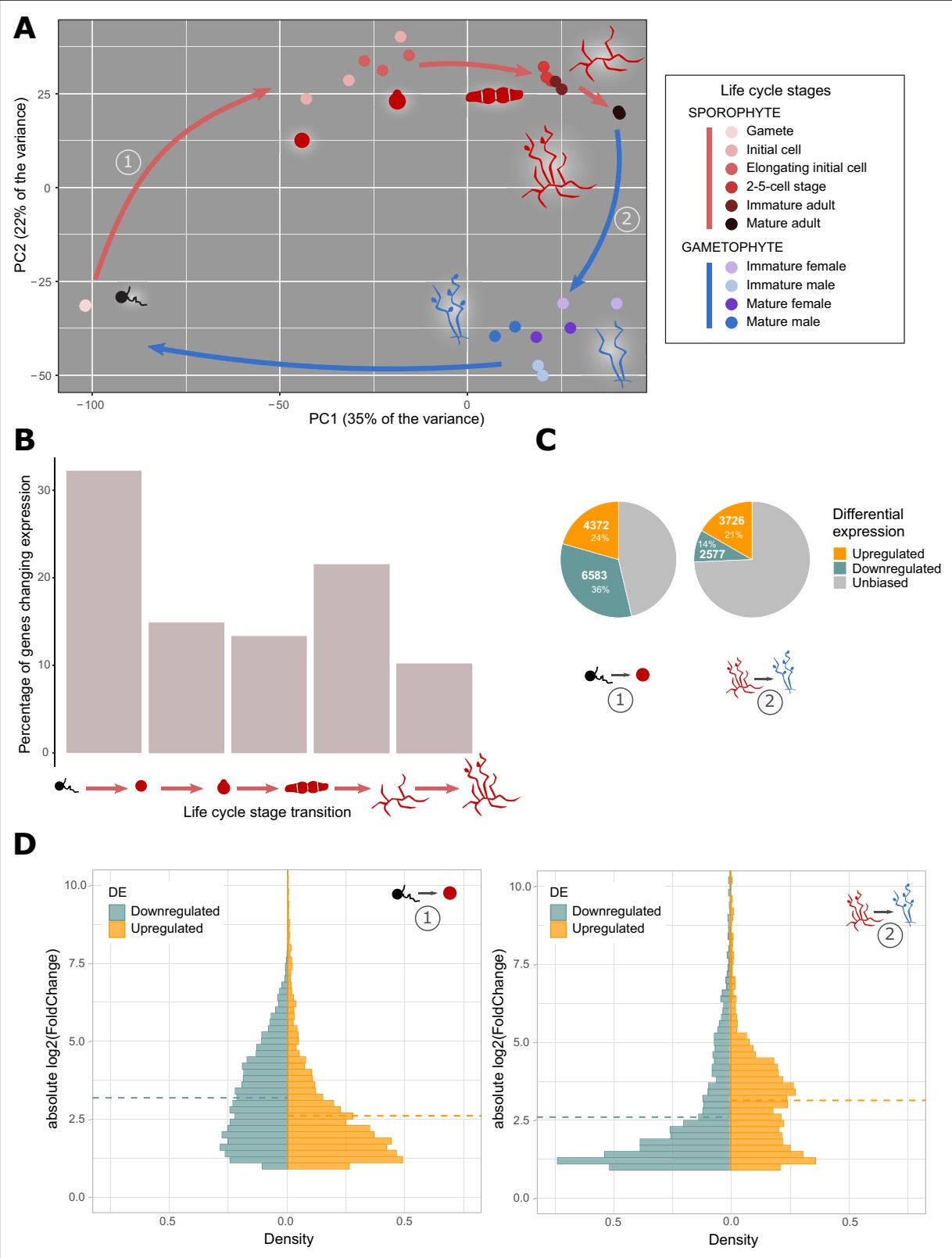

**Figure 3.** Gene expression patterns during the *Ectocarpus* species 7 life cycle. (**A**) Principal component analysis of the *Ectocarpus* species 7 gene expression across 10 life cycle stages. The numbers in circles refer to the differential gene expression analyses illustrated in C and D. Most of the stages analysed consisted entirely of non-flagellated cells with the exception of the gamete stage and mature sporophyte and gametophyte stages, which bear flagellated spores and gametes, respectively, in sporangia and gametangia, respectively. (**B**) Percentage changes in gene expression (regression

*Figure 3 continued on next page*

*Figure 3 continued*

breakpoints) during *Ectocarpus* species 7 sporophyte development (from left to right, transitions between free-swimming male gamete, sporophyte initial cell (24 hours after gamete release), elongating sporophyte initial cell (48 hours after gamete release), sporophyte 2–5 cell stage, non-fertile adult sporophyte and fertile sporophyte). (C) Proportions of differentially expressed genes in the transitions between (1) free-swimming male gamete and sporophyte initial cell stage, and (2) adult sporophyte and gametophyte stages, determined using DESeq2 (*Supplementary file 1*). (D) Density distribution of the |log₂(FoldChange)| for upregulated and downregulated genes in the differential gene expression analysis described in C. Dotted lines indicate mean values.

The online version of this article includes the following figure supplement(s) for figure 3:

**Figure supplement 1.** Light microscopy images of *Ectocarpus* species 7 initial cell and elongating initial cell.

**Figure supplement 2.** Breadth of expression (tau) of genes that were differentially expressed, or not, across the transition from gamete to sporophyte initial cell.

their patterns of expression across the 10 life cycle stages. Using a cut-off of normalised counts ⩾45 (DESeq2 analysis), 16,077 of the 18,370 *Ectocarpus* species 7 genes were classed as being significantly expressed at at least one stage of the life cycle. Of these genes, 14,277 could be assigned to one of 23 gene expression modules, which varied in size from 42 to 2414 genes (*Figure 4—figure supplement 4*, *Figure 4*, *Supplementary file 6*). The modules were given arbitrary colour names (*Figure 4F*). The 24th 'grey' module groups the 1800 genes that were not assigned to a co-expression module (*Figure 4F*, *Supplementary file 6*).

The 23 gene co-expression modules were analysed for enriched gene ontology (GO) terms (*Figure 5* and *Figure 4—figure supplement 2*). For 11 of the 23 gene co-expression modules, the set of enriched GO terms was sufficiently related to allow a general biological function to be manually assigned to the module (*Figure 4A*). These modules indicated that at least a proportion of the genes involved in several cellular processes (photosynthesis, DNA modification, flagella biosynthesis/function, transcription, and translation) are co-ordinately regulated during the life cycle.

The PCA and differential expression analyses described above provided evidence for a large-scale modification of the transcriptome during the transition between the gamete and sporophyte initial cell stage. As expected, changes in expression level at this transition strongly influenced the grouping of genes into expression modules (*Figure 4F*) and several modules were enriched in genes that had been identified as differentially expressed at this first transition (*Figure 4C*). The following two sections will focus on these modules to further characterise this important early developmental transition.

## Sporophyte down-regulated gene modules highlighted DNA- and flagella-related functions in gametes

Seven modules were significantly enriched in genes that were downregulated during the transition from gametes to the sporophyte initial cell (*Figure 4C*, *Supplementary file 6*). For three of these modules, 'pink', 'salmon', and 'turquoise', the genes were also upregulated during the gametophyte stages (*Figure 4D*). The 'pink' module was enriched in gene ontology terms related to nucleosome assembly (*Figure 4—figure supplement 2*) and the 'salmon' module, which contained a large number of monoexonic genes, was enriched in genes predicted to have DNA-related functions (enriched gene ontology terms included 'DNA integration', 'DNA recombination', 'DNA repair', 'DNA helicase', and 'telomere maintenance'; *Figure 4—figure supplement 2*). The 'turquoise' module, which had an expression profile very similar to that of the 'salmon' module (*Figure 4D*), was enriched in genes with predicted functions related to the flagella (enriched gene ontology terms included 'dynein complex', 'cilium', 'BBSome', 'microtubule-based movement', 'axonemal dynein complex assembly', 'cilium movement', and 'microtubule motor activity'), and therefore presumably important for gamete motility (*Figure 4—figure supplement 2*). The upregulation of 'turquoise' module genes in gametophytes is probably explained by the production of flagellated gametes in the gametangia of fertile thalli, although it is not clear why the genes were also expressed at quite a high level in pre-fertile, adult gametophytes. It is possible that transcripts already accumulate at this stage to prepare for the production of large numbers of gametes at fertility or that some of these genes are also involved in other cellular functions, for example related to the cytoskeleton.

Cell-wall biosynthesis functions were not significantly enriched in the 'turquoise', 'salmon', and 'pink' modules, but these modules contained the majority of annotated genes from two families of carbohydrate-active enzymes (CAZymes), the GT23 family (four genes in the 'turquoise' and two genes

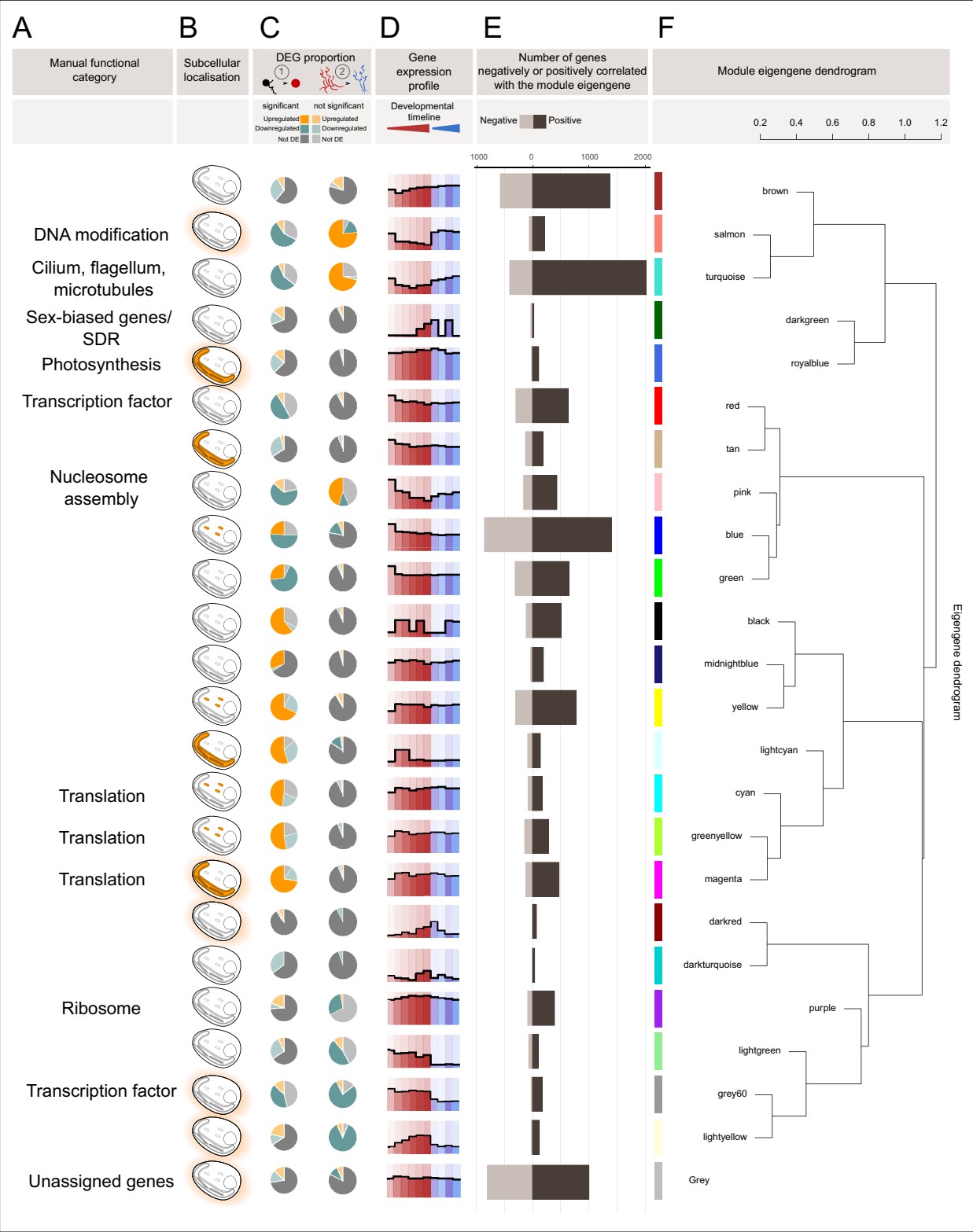

**Figure 4.** Characterisation of gene co-expression modules in *Ectocarpus* species 7. (**A**) Functional categories manually assigned to modules based on Gene Ontology term enrichment analysis of each module. (**B**) Enrichments in subcellular localisations based on HECTAR predictions: mitochondria (small ellipses), plastid (tube shape with thylakoids) and signal peptide/anchor peptide for secreted proteins. Note that HECTAR does not predict nuclear localisation. (**C**) Proportion of differentially expressed genes (DESeq2 adjusted p-value<0.05 and |log2FoldChange|>1) in each gene module in the transitions between (1) free-swimming male gametes and sporophyte initial cell stage, and (2) adult sporophyte and gametophyte (***Supplementary***

*Figure 4 continued on next page*

*Figure 4 continued*

*file 1*). Significant enrichment (indicated by a darker shade of green or orange) means that the proportion of differentially expressed genes in the module was significantly greater (ClusterProfiler adjusted p-value<0.05) than the proportion for the entire genome. (**D**) Average module gene expression profile computed on genes with a WGCNA module MM >0.86 for, from left to right: free-swimming male gamete, sporophyte initial cell (24 hr after gamete release), elongating sporophyte initial cell (48 hr after gamete release), sporophyte 2–5 cell stage, non-fertile adult sporophyte, fertile sporophyte, non-fertile female and male gametophytes, fertile female and male gametophytes developmental stages. (**E**) Number of genes in the module positively or negatively correlated with the module eigengene. (**F**) Module eigengene dendrogram showing the relationship between module eigengenes.

The online version of this article includes the following figure supplement(s) for figure 4:

**Figure supplement 1.** Hierarchical cluster trees showing co-expression modules identified using WGCNA.

**Figure supplement 2.** Enriched GO terms in *Ectocarpus* species 7 gene modules.

**Figure supplement 3.** Transcript abundance heatmap for *Ectocarpus* species 7 CAZYme genes.

**Figure supplement 4.** Transcript abundance heatmap for *Ectocarpus* species 7 transcription factor genes that have one-to-one *D. dichotoma* orthologues.

in the 'salmon' module out of 10 genes in total) and the PL41 family (nine genes in the 'turquoise' module, five genes in the 'salmon' module and three genes in the 'pink' module out of 22 expressed, annotated PL41 genes; *Figure 4—figure supplement 3*). The GT23 family is predicted to encode fucosyltransferases, which would transfer fucosyl groups onto chitin oligosaccharides or glycoproteins, and the PL41 family contains mannose-specific alginate lyases (*Inoue and Ojima, 2019*; *Mazéas et al., 2024*). While the role of GT23 enzymes in gametes is unclear, it is possible that the polysaccharide depolymerisation action of PL41 lyases may facilitate gamete release from plurilocular sporangia in a manner analogous to the actions of pectate lyases and rhamnogalacturonan lyases during fruit ripening in land plants (*Méndez-Yañez et al., 2020*; *Uluisik and Seymour, 2020*; *Al-Hinai et al., 2024*).

The remaining four modules that were significantly enriched in genes downregulated during the transition from the gamete to the early sporophyte ('blue', 'green', 'red', and 'grey60') did not exhibit gene upregulation during the gametophyte generation (*Figure 4C and D*, *Supplementary file 6*). No significant GO term enrichment was detected for the 'blue' and 'green' modules. The 'red' and 'grey60' modules were enriched in transcription factors and related GO terms such as 'DNA-binding domain' (*Figure 4—figure supplement 2*). To investigate this point further, a recently established dataset of 325 transcription-associated proteins (TAPs) for *Ectocarpus* species 7 (*Denoeud et al., 2024*) was enriched with the 90 Esv-1–7 domain proteins (*Macaisne et al., 2017*) from this species (*Supplementary file 5*, *Figure 4—figure supplement 4*), as the latter have been hypothesised to function as zinc finger transcription factors (*Macaisne et al., 2017*), and the expression of these genes analysed in gametes. Transcripts of approximately a third of the genes (147) in this transcription-associated dataset were more abundant in gametes than at the sporophyte initial cell stage (24 hr) but the proportion was similar to the proportion for all the genes in the genome indicating that there was not an overall enrichment of the gamete transcriptome in transcription-associated genes.

Taken together, analysis of the seven modules highlighted the importance of DNA-related functions, including nucleosome assembly and transcription, and flagella-related functions in gametes.

## Subsets of translation-related genes are upregulated in the sporophyte initial cell but the gamete already expresses some translation-related and photosynthesis genes

Nine modules were significantly enriched in genes upregulated during the transition between gametes and the sporophyte initial cell (*Figure 4C*, *Supplementary file 6*). Most of the nine modules could not be clearly associated with a general cellular process, but three ('cyan', 'magenta', and 'greenyellow') were enriched in functions related to translation (*Figures 4A and 5*). These three modules exhibited very similar expression profiles (*Figure 5A*) but were enriched in different classes of translation-related gene (*Figure 5B–E*). The 'greenyellow' module contained the RNA polymerase I gene which synthesises rRNA and was enriched in GO terms related to 'nucleolus', 'small-subunit processome', 'ribosome large subunit assembly', 'ribosome biogenesis', and 'rRNA processing'. This module was therefore essentially related to the biosynthesis of the ribonucleic component of the

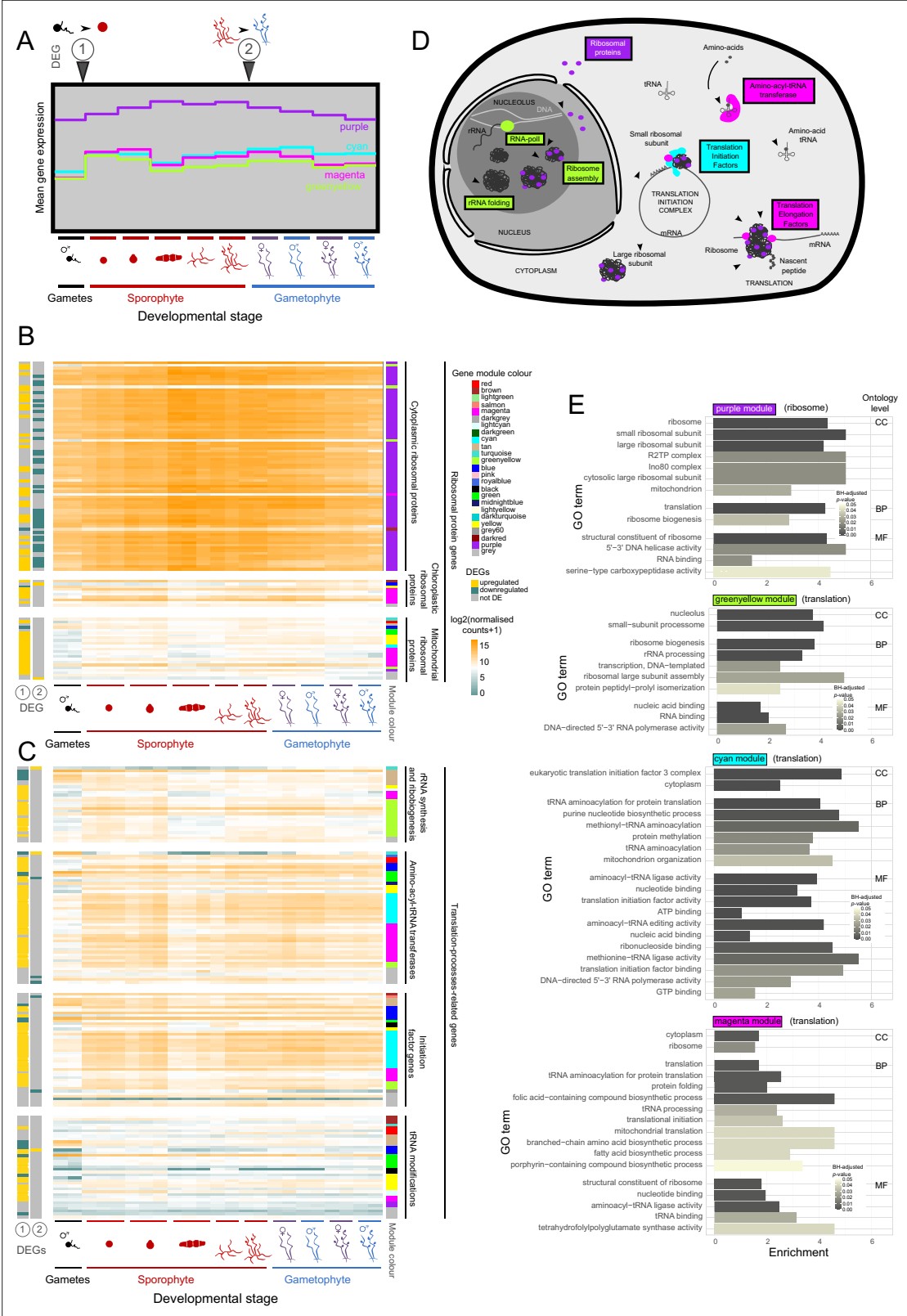

**Figure 5.** Expression of translation-related genes during *Ectocarpus* species 7 development. (**A**) Mean expression profile computed for genes with a WGCNA module MM >0.86 for (from left to right): free-swimming male gamete, sporophyte initial cell (24 hr after gamete release), elongating sporophyte initial cell (48 hr after gamete release), sporophyte 2–5 cell stage, non-fertile adult sporophyte, fertile sporophyte, non-fertile female and male gametophyte, fertile female and male gametophyte developmental stages in modules 'purple', 'cyan', 'greenyellow', 'magenta'. (**B and**

*Figure 5 continued on next page*

*Figure 5 continued*

**C**) Heatmaps showing $\log_2$(NormalisedCounts +1) values across the same developmental timepoints as in **A** for manually reannotated ribosomal protein genes corresponding to cytosolic, chloroplast and mitochondrial ribosomal subunits (**B**) and translation-related genes (**C**). Left annotation track: differential expression analysis results for the transitions between (1) free-swimming male gamete and sporophyte initial cell stage, and (2) adult sporophyte and adult gametophyte; right annotation track: WGCNA module colours. (**D**) Schematic representation of translation-related functions enriched in the 'greenyellow', 'purple', 'magenta', 'cyan' modules. When it was possible to manually assign a general function to a module, the annotation is indicated in brackets after the module name. (**E**) Gene ontology terms significantly enriched in the sets of genes clustered within the 'purple', 'greenyellow', 'cyan', and 'magenta' modules in *Ectocarpus* species 7. Enrichment is indicated as log2 of the ratio of the proportion of genes assigned to the GO term in the module divided by the proportion for the whole genome. CC, cellular component; BP, biological process; MF, molecular function.

The online version of this article includes the following figure supplement(s) for figure 5:

**Figure supplement 1.** Transcript abundance heatmap for *Ectocarpus* species 7 ribosomal genes.

**Figure supplement 2.** Three-dimensional representations of a ribosome illustrating differential expression of ribosomal protein genes.

ribosome (*Figure 5D*). The 'cyan' and 'magenta' modules were enriched in GO terms such as 'tRNA aminoacylation for protein translation', 'tRNA processing', 'translation initiation' (*Figure 5E*). These two clusters included 25 of the 49 aminoacyl-tRNA-transferases annotated in the *Ectocarpus* species 7 genome, and also included a number of genes related to transcription initiation (*Figure 5C*). Notably, the 'cyan' module included nine eIF3-subunit-encoding genes out of a total of 15 annotated genes in the genome (*Figure 5C*). Interestingly, the 'cyan' and 'greenyellow' modules were enriched (p<0.05 clusterProfiler) in genes with a mitochondrial targeting sequence (*Figure 4B*), whereas the 'magenta' module was enriched (p<0.05 clusterProfiler) in genes that encode proteins targeted to the plastid. The plastid-targeted proteins had predicted functions other than photosynthesis (e.g. translation and transport).

The 'cyan', 'greenyellow', and 'magenta' modules clustered together based on their average expression profiles (*Figure 4F*), suggesting that the three groups of translation-related functions share some level of co-expression. Taken together, the gene content of these three clusters suggests that there is a marked increase in translation in the initial cell compared to the gamete, both in the cytosolic and the organellar compartments. Importantly, however, not all translation-related genes exhibited marked upregulation during the transition from the gamete to the initial cell stage. For example, the transcripts of the many ribosomal-protein-encoding genes that were grouped in the 'purple' module were already present in gametes at relatively high levels (*Figure 5B*, *Figure 5—figure supplement 1*). The ribosomal protein genes are an interesting case because, whilst the majority of these genes were assigned to the 'purple' module, corresponding essentially to constitutive expression, a minority of the members of this broad family exhibited more complex expression patterns and were assigned to other modules (*Figure 5B*, *Figure 5—figure supplement 1*). In general, this latter minority of genes tends to encode proteins that are located in close vicinity in the P-stalk within the ribosome (*Figure 5—figure supplement 2*). The functional significance of this clustering is unclear, but it has been proposed that the P-stalk influences translation fidelity and the pool of mRNAs associated with the ribosome (*Wawiórka et al., 2017*; *Dopler et al., 2024*). We also noted that ribosomal protein genes with non-constitutive expression patterns tended to have paralogues that were constitutively expressed (*Figure 5—figure supplement 1*). Interestingly, cytosolic and organellar ribosomal protein genes did not exhibit the same patterns of expression with respect to the life cycle, with the genes encoding organellar proteins exhibiting more complex expression patterns (i.e. they were assigned to multiple modules; *Figure 5B*).

The 'royalblue' module is another example of a module which did not exhibit marked changes in expression between the gamete and the sporophyte initial cell. This module was highly enriched in genes related to photosynthesis or encoding chloroplast-targeted proteins (*Figure 4A, B and C*). Based on the transcriptomic data, therefore, both cytosolic translation and photosynthesis appear to be processes that are already important at the gamete stage, although it is possible that transcripts related to these processes may accumulate in gametes but be translated at a later stage.

## Comparative analysis of life-cycle-related gene expression between *Ectocarpus* species 7 and *D. dichotoma*

To determine whether the developmental and life-cycle-related transcription patterns observed in *Ectocarpus* species 7 corresponded to common, fundamental features that are shared with other brown algae, a comparative analysis was carried out between *Ectocarpus* species 7 and *D. dichotoma*. These two species are phylogenetically distant within the brown algal tree (*Akita et al., 2022*). Based on a recent estimate (*Choi et al., 2024*), their common ancestor dates to about 250 Mya. Consequently, if a feature is shared between the two species, this is an indication that it may have been conserved over a significant part of the evolutionary history of the brown algal lineage.

Replicated transcriptomic data was available for seven stages of the *D. dichotoma* life cycle (*Supplementary file 1*). In a PCA analysis (*Figure 6—figure supplement 1*), the adult sporophyte and gametophyte stages clustered together, presumably reflecting the isomorphy between the two generations. Consequently, the distribution of stages in the PCA did not follow the progression of the life cycle, as had been observed for *Ectocarpus* species 7. However, the first dimension clearly separated male gametes from the other *D. dichotoma* life cycle stages, and the second dimension separated early developmental stages from adult stages. *D. dichotoma* has flagellated male gametes and non-flagellated egg cells. The latter were clustered with early sporophyte stages (zygote and 8 hr embryo), suggesting that the markedly different gene expression patterns observed in male gametes from the two species compared to other life cycle stages were related to their being actively swimming, flagellated cells rather than being a universal characteristic of all brown algal gametes.

Life-cycle-related gene expression in *Ectocarpus* species 7 and *D. dichotoma* was compared using two different approaches. The first aimed to determine whether the co-expression patterns of *Ectocarpus* species 7 genes grouped into modules were mirrored by the *D. dichotoma* orthologues of these genes. The second involved independently clustering *D. dichotoma* genes into co-expression modules using life cycle transcriptomic data and then determining whether there were similarities between the modules established for the two species.

To optimally identify conserved one-to-one orthologues between *Ectocarpus* species 7 and *D. dichotoma*, an orthologue analysis was carried out specifically for the two species using Orthofinder (*Emms and Kelly, 2019*). This approach identified 8527 one-to-one orthologous relationships between the 18,370 *Ectocarpus* species 7 genes and the 20,583 *D. dichotoma* genes (*Supplementary file 7*). This orthologue dataset was used to compare gene expression patterns in the two species.

## *D. dichotoma* orthologues of sets of genes from *Ectocarpus* species 7 modules exhibited conserved co-expression for some modules that grouped genes involved in the same cellular process

To determine whether the co-expression patterns of genes in *Ectocarpus* species 7 modules were conserved in *D. dichotoma*, the *Ectocarpus* species 7 genes in each module were substituted with their *D. dichotoma* orthologues and various parameters, notably the preservation of the density of the module and the preservation of the connectivity were evaluated (*Supplementary file 8*) using the *D. dichotoma* transcriptomic data for the seven different life cycle stages (*Supplementary file 1*). The density preservation of a module was measured by evaluating whether the average correlation for the module was conserved in the second species and whether the module summary profile (module eigengene) still explained a high proportion of the variance after the genes were substituted with orthologues from the second species. Connectivity preservation, on the other hand, determined whether the correlation pattern between the genes was conserved after substitution. Various density and connectivity statistics based on functions in both the WGCNA R package and the NetRep R package were aggregated to estimate conservation of gene co-expression for each *Ectocarpus* species 7 module in *D. dichotoma*.

Generally, the modules with the best conservation statistics in this analysis were those that had been identified as being highly enriched in *Ectocarpus* species 7 genes with predicted housekeeping functions (*Figure 6A*). Thus, the 'photosynthesis' ('royalblue'), the 'ribosome' ('purple'), the 'cilium and flagellum' ('turquoise'), and various translation-associated ('greenyellow', 'cyan', and 'magenta') modules all showed a high level of conservation. The expression pattern of the 'darkgreen' module was also considered to be conserved in *D. dichotoma* despite the fact that only three orthologues were detected. The 'dark green' module consists principally of genes located within the non-recombining

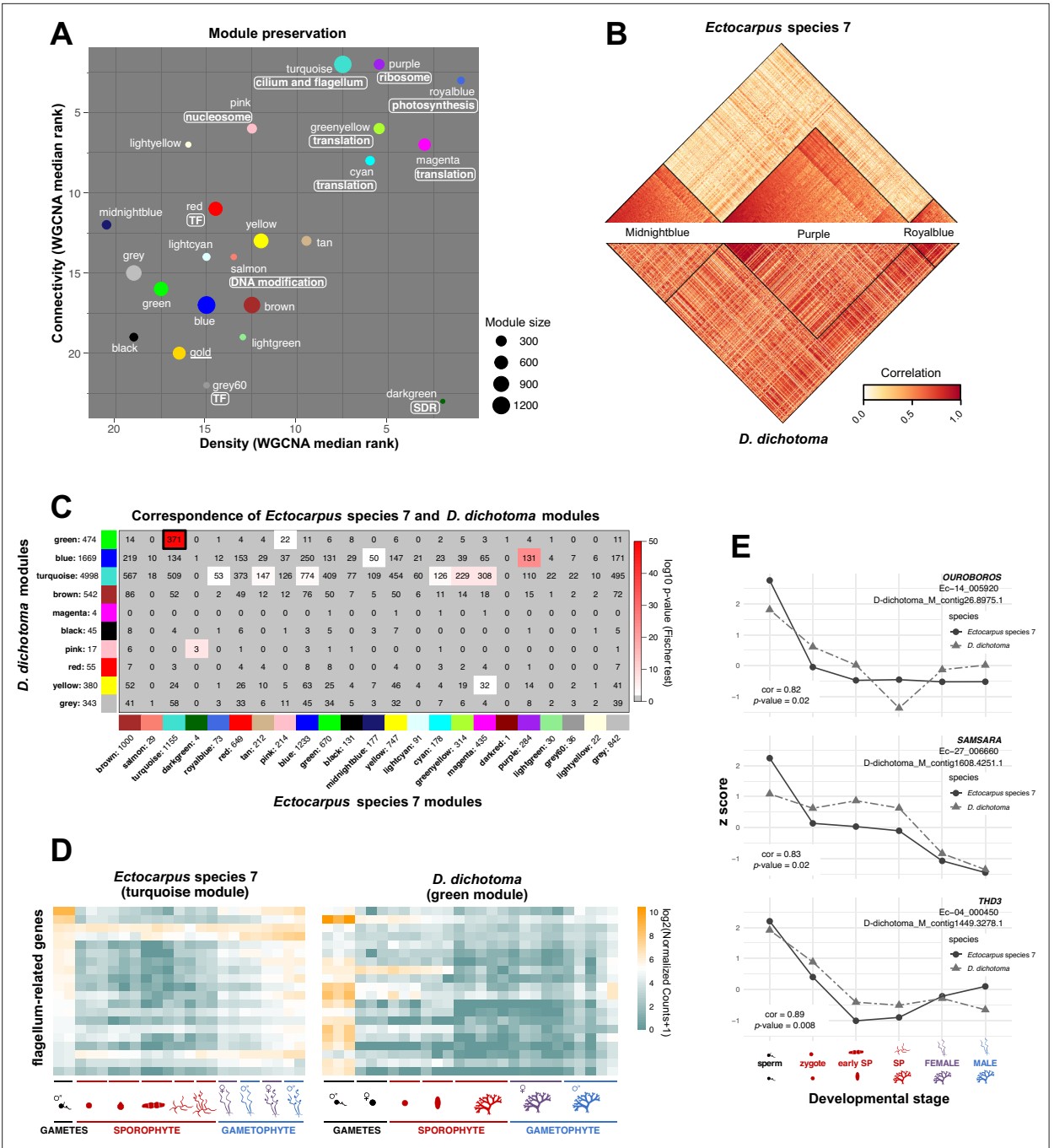

**Figure 6.** Conservation of life-cycle-related gene co-expression patterns between *Ectocarpus* species 7 and *D. dichotoma*. (**A**) WGCNA density and connectivity median rank statistics indicating the degree of conservation of gene co-expression patterns when *Ectocarpus* species 7 genes in *Ectocarpus* species 7 co-expression modules were substituted with their *D. dichotoma* orthologues. Manually assigned general biological functions are indicated in boxes. SDR, sex-determining region. Module size indicates the number of one-to-one orthologues in each module. The modules 'darkred' and 'darkturquoise' were not included in this analysis because they had only one and zero orthologues, respectively (***Supplementary file 6***). TF, transcription factor; SDR, sex-determining region. (**B**) Correlation heatmap comparing three *Ectocarpus* species 7 modules between *Ectocarpus* species 7 and *D. dichotoma*: 'midnightblue' (very poor conservation), 'purple', and 'royalblue' (good conservation). The lower half of the heatmap was calculated based on the expression pattern of the *D. dichotoma* orthologue of each *Ectocarpus* species 7 gene in each module. (**C**) Counts of shared one-to-one orthologues between gene co-expression modules defined for *Ectocarpus* species 7 (x-axis) and *D. dichotoma* (y-axis). The colour code represents the $\log_{10}(p\text{-value})$ for Fisher's exact test (red bar), which was applied to determine whether pairs of modules shared a greater number of one-to-one orthologues than expected from a random distribution. Numbers after the module names indicate the number of one-to-one orthologues in each module. The 'dark turquoise' module was not included in this analysis because it contained zero orthologues (***Supplementary file 6***). (**D**) Heatmap

*Figure 6 continued on next page*

*Figure 6 continued*

showing the expression levels (log$_2$(NormalisedCounts +1)) of 20 selected one-to-one orthologous flagellum-related genes in *Ectocarpus* species 7 and *D. dichotoma*. *Ectocarpus* species 7 timepoints: free-swimming male gamete, sporophyte initial cell (24 hr after gamete release), elongating sporophyte initial cell (48 hr after gamete release), sporophyte 2–5 cell stage, non-fertile adult sporophyte, fertile sporophyte, non-fertile female and male gametophytes, fertile female and male gametophytes. *D. dichotoma* timepoints: sperm, egg cell, zygote, embryo, adult non-fertile sporophytes, and female and male gametophytes. (**E**) Computed z score based on log$_2$(NormalisedCounts +1) values for three TALE homeodomain transcription factors over six developmental stages shared between *Ectocarpus* species 7 and *D. dichotoma*, namely sperm, zygote, early sporophyte, non-fertile sporophyte, female and male gametophyte. The Pearson correlation coefficient between the two expression datasets (cor) and the associated p-value is indicated.

The online version of this article includes the following figure supplement(s) for figure 6:

**Figure supplement 1.** *D. dichotoma* PCA and modules.

**Figure supplement 2.** Enriched GO terms *D. dichotoma* gene modules.

**Figure supplement 3.** Transcript abundance heatmap for one-to-one orthologous transcription factors between *Ectocarpus* species 7 and *D. dichotoma*.

sex-determining region (SDR) of the female (U) and male (V) sex chromosomes (*Ahmed et al., 2014*) and is therefore a special case because the co-expression pattern is determined not only by gene expression but also by gene presence (i.e. presence of the U or V sex chromosome in gametophytes). The module was considered to be conserved in *D. dichotoma* because the three orthologues are located in the SDR in both species.

For modules that had good conservation statistics between the two species, *D. dichotoma* orthologues of the *Ectocarpus* species 7 genes that were best correlated with the module eigengene also exhibited a strong correlation with the module eigengene after the *Ectocarpus* species 7 genes had been replaced with the *D. dichotoma* orthologues (*Figure 6B*). For example, for the 'purple' and 'royalblue' modules (which had good conservation statistics; *Figure 6A*) in the top part of *Figure 6B*, the *Ectocarpus* species 7 genes that were best correlated with the module eigengene are positioned on the left. In the bottom part of the figure, the *D. dichotoma* orthologues of these genes are also positioned on the left and they also show strong correlation with the module eigengene. This conservation of correlation with the module eigengene between orthologues was not observed for modules such as 'midnightblue' that had poor conservation statistics between the two species (*Figure 6B*).

To summarise, this analysis indicated that the co-expression pattern of the genes in an *Ectocarpus* species 7 module following replacement with the *D. dichotoma* orthologues was only conserved for a subset of the modules. The modules that exhibited conservation of co-expression across the two species in this analysis tended to contain sets of genes that functioned in a basic, evolutionarily-conserved cellular process. Moreover, conservation of module co-expression patterns across the two species appeared to be conferred preferentially by orthologues of the *Ectocarpus* species 7 genes that showed the best correlation with the module eigengene.

## A subset of life-cycle-related gene co-expression modules was conserved between *Ectocarpus* species 7 and *D. dichotoma*

For the second approach to compare life-cycle-regulated gene expression in *Ectocarpus* species 7 and *D. dichotoma,* WGCNA was used to define co-expressed gene modules independently for *D. dichotoma* using the replicated transcriptomic data for the seven different *D. dichotoma* life cycle stages (*Supplementary file 1*; *Figure 4—figure supplement 4B*). The number of modules defined for *D. dichotoma* was lower than the number that had been obtained for *Ectocarpus* species 7 (nine compared with 23; *Figure 6—figure supplement 1*, *Supplementary file 6*, *Supplementary file 9*, *Supplementary file 10*), presumably because data was available for fewer life cycle stages and because the *D. dichotoma* adult sporophyte and gametophyte generations are isomorphic and therefore have more similar overall gene expression patterns (*Figure 6—figure supplement 1A*). As for the *Ectocarpus* species 7 modules, analysis of enriched GO terms (*Figure 6—figure supplement 2*) allowed general biological functions to be manually assigned to a subset of the modules (five of the nine modules; *Figure 6—figure supplement 1A*).

Correlation between modules across the two species was assessed based on the number of shared one-to-one orthologues in pairwise comparisons. Fisher's exact test was used to determine whether *Ectocarpus* species 7 and *D. dichotoma* modules shared a greater number of one-to-one orthologues

than expected from a random distribution. This analysis identified several pairs of life-cycle-related gene co-expression modules with significantly conserved gene contents between the two species, but not all modules were conserved (*Figure 6C*). As in the first analysis described above, conservation tended to be strongest for *Ectocarpus* species 7 modules that had been manually assigned general functional annotations and which had registered the highest values for module preservation statistics in the first analysis (*Figure 6A*).

'Purple' was one of the most strongly conserved *Ectocarpus* species 7 modules: 131 of its 284 one-to-one orthologues were shared with the *D. dichotoma* 'blue' module (*Figure 6C*), with most of the shared orthologues predicted to encode ribosomal proteins. However, the *D. dichotoma* 'blue' module exhibited a marked downregulation in male gametes, which was not observed for the *Ectocarpus* species 7 'purple' module. In contrast, the *Ectocarpus* species 7 'turquoise' module shared 371 (of a total of 1155) one-to-one orthologues with the *D. dichotoma* 'green' module, principally with predicted cilium- and flagellum-related functions (*Figure 6C and D*), and these two modules also exhibited similar life-cycle-related expression patterns, in particular high levels of expression in male gametes. Therefore, both the gene content and the expression patterns of these two modules appear to have been conserved over evolution. Most of the one-to-one orthologues in the translation-related *Ectocarpus* species 7 modules 'cyan', 'greenyellow', and 'magenta' corresponded with genes in the *D. dichotoma* 'turquoise' module, consistent with the poorer resolution of the *D. dichotoma* modules (*Figure 6C*). Nonetheless, all these modules exhibited a lower level of gene expression in male gametes, indicating conservation of their expression patterns over evolutionary time.

Taken together, the comparison between *Ectocarpus* species 7 and *D. dichotoma* indicated that there has been partial conservation of life-cycle-related gene expression patterns between the two species, in particular for modules corresponding to clearly identifiable cellular functions. Moreover, some modules exhibited conservation not only in terms of gene (orthologue) content but also exhibited similar expression patterns over the course of the life cycle, proving further support for the (partial) conservation of life-cycle-related gene expression patterns over time.

The above analyses provided no evidence that transcription-related modules were conserved, overall, between *Ectocarpus* species 7 and *D. dichotoma*, but a more detailed analysis was carried out to determine whether life-cycle-related expression patterns might have been conserved for individual transcription factor genes. For this, 214 transcription factors with one-to-one orthologues in the two species were selected and their expression patterns (normalised expression z-scores) analysed across six life cycle stages that were approximately equivalent between the two species (gamete, zygote, early sporophyte, adult sporophyte, adult male gametophyte, and adult female gametophyte; *Figure 6—figure supplement 3*, *Supplementary file 1*). High Pearson correlation coefficients were obtained for a subset of the transcription factors, but none of the correlations were significant at $p=0.05$ after correction for multiple testing (*Supplementary file 11*). However, the applied correction was stringent because a large number of orthologous genes were considered, so the most strongly correlated transcription factors may nonetheless merit further study in the future. We noted, for example, that all three members of the TALE homeodomain transcription factor family, which includes the life cycle regulators *ORO* and *SAM* (*Arun et al., 2019*), were among the genes that exhibited high Pearson correlation coefficients in this analysis (*Supplementary file 11*, *Figure 6E*).

## Discussion

### Subsets of genes with related functions are co-expressed during the life cycle

This study analysed replicated RNA-seq samples for 10 different stages of the *Ectocarpus* species 7 life cycle to provide an overview of life-cycle-related gene expression in this species. Clustering of genes into co-expression modules indicated that subsets of the genes associated with several cellular processes, including transcription, protein synthesis, flagella biosynthesis/function and photosynthesis, are co-ordinately regulated during the life cycle.

The coordinated expression patterns of the genes in these modules imply coordinated regulation of transcription and/or mRNA stability over the course of the life cycle. The regulatory systems that mediate this coordinated expression are currently unknown for brown algae but information about the mechanisms that co-ordinately regulate these processes in other eukaryote lineages can provide

some indications of possible regulators. For example, in *Ectocarpus* species 7, four different modules were enriched in translation-related genes. In other eukaryotic lineages, a large proportion of ribosomal protein genes are regulated by the mTOR/PKA pathway (*Martin et al., 2004*; *Ni and Buszczak, 2023*). Interestingly, genes predicted to encode cytosolic and organellar ribosomal proteins were regulated differently in *Ectocarpus* species 7. Note also that this conservation of expression patterns provides additional support for the transfer of the *Arabidopsis* ribosomal protein gene nomenclature (*Scarpin et al., 2023*) to *Ectocarpus* species 7 genes (https://bioinformatics.psb.ugent.be/orcae/overview/EctsiV2) based on sequence similarity.

In land plants, photosynthesis-related genes have been shown to be regulated by both light and developmental cues and the regulatory pathways have been characterised. Phytochrome signalling counteracts negative regulation of photosynthesis genes by transcription factors such as PHYTOCHROME-INTERACTING FACTOR (PIF), PIF-LIKE (PIL), and GOLDEN2-LIKE1 (GLK1), which act downstream of various developmentally regulated pathways (*Waters et al., 2009*; *Wang et al., 2017*). However, orthologues of these transcription factors have not been found in brown algae and, consequently, coordinated regulation of photosynthetic genes in brown algae, as was observed here for the 'royalblue' module, presumably involves a different set of actors.

As far as flagella- and cilium-related genes are concerned, these organelles are widespread in eukaryotes and the last eukaryotic common ancestor (LECA) probably possessed cilium-related structures (*Carvalho-Santos et al., 2011*). In Metazoans, two transcription factor families, FOXJ1 and RFX, control the formation of core components of all types of cilia (*Choksi et al., 2014*). However, although *Saccharomyces cerevisiae* also possesses an *RFX* gene, this family of transcription factors has only been shown to regulate ciliogenesis in metazoans and is not found outside the opisthokonts (*Chu et al., 2010*). In *Chlamydomonas reinhardtii*, ciliogenesis is under the control of XAP5 (*Li et al., 2018*), a transcription factor that is widely distributed across eukaryotic lineages. Neither *FOXJ1* nor *RFX* has been found in brown algal genomes, but *Ectocarpus* species 7 does possess an *XAP5* homologue (LocusID Ec-05_002220). This gene therefore represents a possible candidate for future analysis of the regulation of this pathway.

These examples illustrate how candidate regulatory genes may be identifiable for cellular processes that are highly conserved across eukaryotes, but candidates are more difficult to identify for modules that group brown-algal-specific genes of unknown function or functions such as the PL41 and GT23 gene families that do not have equivalents in other eukaryotic lineages. In some instances, regulatory genes may show similar expression patterns to the genes they regulate as a result of auto-regulation and therefore be included within a co-expression module, but this will not be the case for all regulators.

## Identification of marked transcriptome changes during early sporophyte development

Analysis of RNA-seq data for multiple developmental stages of *Ectocarpus* species 7 provided a broad overview of gene expression throughout the life cycle, with a particular focus on the early stages of sporophyte development. Progression through the life cycle was correlated with concomitant, progressive changes in the composition of the transcriptome (*Figure 3A*) and a particularly marked change in composition occurred at the transition from the gamete stage to the initial stage of sporophyte development. Most of the gene co-expression modules identified in this study exhibited changes in transcript abundance at this transition, and more than half of the genes in the genome were differentially expressed between the two stages.

Periods of large-scale changes in transcriptome composition have been observed during early development of both animals and land plants (*Xue et al., 2013*; *Anderson et al., 2017*; *Vastenhouw et al., 2019*; *Zhao et al., 2019*), where they tend to coincide with the maternal-to-zygotic transition (MZT). The MZT corresponds to the switch from reliance on maternal transcripts supplied to the transcriptionally inactive egg cell to mRNAs produced by transcription of the zygote genome (*Tadros and Lipshitz, 2009*; *Vastenhouw et al., 2019*). During the MZT, induction of the expression of genes in the zygote genome, which is referred to as zygotic gene activation (ZGA), coincides with a phase of degradation of maternal transcripts in both animals and land plants. The exact timing of MZT varies across species, occurring rapidly in land plants, only hours after zygote formation (*Anderson et al., 2017*; *Chen et al., 2017*; *Zhao et al., 2019*), but considerably later in animals, for example occurring 3 days after fertilisation in eight-cell human embryos (*Xue et al., 2013*).

However, even in animals, one-cell stage embryos exhibit a distinct transcriptome pattern (*Xue et al., 2013*).

*Ectocarpus* is not oogamous and both male and female gametes are small, flagellated cells that swim actively for a significant part of their lifetime. It therefore seems likely that these cells are already transcriptionally active during the gamete stage, in which case they would not rely solely on maternally produced transcripts (although this has not yet been investigated experimentally). Consequently, it is not clear whether there is a specific phase of degradation of maternal transcripts in gametes equivalent to the phases of maternal transcript degradation that have been observed at the MZT in land plants and animals. However, the marked change in transcriptome composition during early development is a feature that is shared with animals and land plants, underlining the broad importance of large-scale transcriptional changes during early development.

In terms of functional categories, fewer genes involved in flagellar function or transcription were expressed after the transition from the gamete stage to the initial stage of sporophyte development. Downregulation of flagellar genes is expected for this transition from a flagellated to a non-flagellated cell. The reduction in the number of expressed transcription factor genes is more surprising, but it is likely that the nature of the transcription factor genes expressed or repressed at this stage is more important than the overall number of transcription factor genes expressed. Trends in transcription factor gene up- and down-regulation are not consistent across species, with a decrease in the number of different transcription factor gene transcripts observed in *Arabidopsis* (*Zhao et al., 2019*) but an increase in rice (*Anderson et al., 2017*), maize (*Chen et al., 2017*), and human (*Xue et al., 2013*), coinciding with the MZT. Moreover, in rice and maize, specific loci that play key roles during embryogenesis such as members of the BABYBOOM and WUSCHEL gene families are upregulated at the MZT/ZGA, supporting the idea that the identity of the specific transcription factor genes expressed is more important than the overall numbers (*Anderson et al., 2017*; *Chen et al., 2017*). Note that *Ectocarpus* has AP2 domain and homeodomain proteins but no close homologues of the BABYBOOM and WUSCHEL families, so it is likely that different transcriptional regulators are involved in regulating development at the equivalent stage in brown algae.

Analysis of modules that exhibited overall gene upregulation between the gamete and early sporophyte stages identified translation as a general function that was upregulated during this transition. However, the upregulation of these genes does not appear to have corresponded to de novo induction of translation because a subset of translation-related genes was already expressed at the gamete stage. The overall gene expression pattern is more indicative of a step increase in translation capacity or complexity in the early sporophyte compared to the gamete.

## Conservation of life-cycle-related gene expression patterns across brown algal species

Comparison of generation-biased gene expression between adult gametophyte and sporophyte stages across 10 brown algal species identified a strong positive correlation between the degree of dimorphism between generations and the number of GBGs identified per species, indicating that the majority of GBG expression is related directly or indirectly to thallus morphology, at least for the adult stages analysed. Orthologue analysis indicated only weak overlap between GBG sets from different species, supporting an earlier conclusion that there is rapid turnover of GBG sets in brown algae (*Lipinska et al., 2019*). Interestingly, however, analysis of GO enrichment in GBG sets indicated that several broad functional categories were conserved across gametophyte-biased gene sets, whereas different broad functional categories were conserved across sporophyte-biased gene sets (*Figure 2*). This observation suggests that the two generations have conserved life-cycle-related functional characteristics that are independent of morphological complexity or the degree of dimorphism between generations.

The analysis of whole adult individuals to study GBG expression therefore identified some interesting conserved life-cycle-related features, but the resolution of this approach is limited because only one sporophyte stage and one gametophyte stage was analysed per species and because, at the adult stage, the data represent an average view of the transcriptomes of multiple different cell types. The comparative analysis of multiple stages of early sporophyte development in *Ectocarpus* species 7 and *D. dichotoma* allowed a more detailed analysis of developmentally regulated gene expression and, notably, showed that there was considerable conservation of co-expressed gene modules

between two species, particularly for modules that were enriched in genes involved in evolutionarily-conserved functional categories. Moreover, when the known life cycle regulatory genes *SAM* and *ORO* were analysed, there was clear evidence of conservation of their expression patterns across the two species. These results underline the importance of carrying out detailed transcriptomic analyses to identify and investigate conserved life cycle regulatory pathways.

Neither *Ectocarpus* species 7 nor *D. dichotoma* exhibits a high degree of dimorphism between life cycle generations. It would be of interest in the future to carry out a similar analysis to those carried out here for a species with highly dimorphic generations such as the kelp *Saccharina latissima*.

## Materials and methods
### Life cycle stages and variants on the sexual cycle
Both *Ectocarpus* species 7 and *D. dichotoma* have haploid-diploid life cycles that involve alternation between a sporophyte and a gametophyte generation (*Müller, 1967*; *Hoek et al., 1995*). During the sexual life cycle, the diploid sporophyte generation produces haploid meio-spores through meiosis. These spores germinate into haploid gametophytes that can be male or female (dioicy). Male and female gametes, produced, respectively, by male and female gametophytes, fuse to produce a diploid zygote, which, in turn, develops into the next sporophyte generation. The zoids (gametes and spores) are the only flagellated stages of the life cycle. In *Ectocarpus* species 7, the germinating zygote undergoes a developmentally symmetrical first division to produce a basal filament, which then branches to establish a basal filament system consisting of round and elongated cells. Once this basal system is established, upright filaments consisting of cylindrical cells grow up into the medium and these filaments bear the sexual structures, plurilocular sporangia and unilocular sporangia (*Peters et al., 2008*). *Ectocarpus* species 7 also exhibits variations on this basic sexual life cycle, at least in culture (*Müller, 1967*). For example, parthenogenesis of unfertilised gametes gives rise to haploid partheno-sporophytes. A few hours after release from the plurilocular sporangia, unfertilised free-swimming male gametes attach to the substrate, lose their flagella and acquire a round shape. Upon attachment, they start to develop a cell wall. After about 2 days, the initial cell elongates and divides, producing the first cells of the sporophytic basal filaments. All but one of the sporophyte stages analysed in this study corresponded to partheno-sporophyte samples. Partheno-sporophytes exhibit identical morphology and reproductive functionality to diploid sporophytes, and pure samples of early-stage diploid sporophytes are difficult to obtain because of a high level of gamete parthenogenesis in genetic crosses (*Coelho et al., 2012a*).

### Publicly available RNA-seq datasets
RNA-seq data for the sporophyte and gametophyte generations of the following 10 species was recovered from the Phaeoexplorer website (https://phaeoexplorer.sb-roscoff.fr/) or from public databases: *Macrocystis pyrifera*, *Saccharina latissima*, *Saccharina japonica*, *Saccorhiza polyschides*, *Myriotrichia claveaformis*, *Pylaiella littoralis*, *D. dichotoma*, *Sphacelaria rigidula*, *Ectocarpus* species 7, and *Scytosiphon promiscuus*. In addition, RNA-seq data was collected for multiple stages of the *D. dichotoma* and *Ectocarpus* species 7 life cycles. All the accession numbers for these data are listed in *Supplementary file 1*.

### Comparisons of orthologues across multiple brown algal species
To compare orthologues across the 10 brown algal species analysed for generation-biased gene expression, an orthogroup analysis was carried out using Orthofinder (*Emms and Kelly, 2019*) version 2.5.2. Strict selection of only one-to-one orthologues did not identify sufficient orthologues (1038), principally due to gene model fragmentation in the lower quality genome assemblies. A more relaxed selection approach was therefore used, taking into account genome assembly quality. This relaxed approach, which required exactly one orthologue from the best quality assemblies (*D. dichotoma*, *Ectocarpus* species 7, *P. littoralis*, *S. latissima*, *S. promiscuus,* and *S. rigidula*), between zero and two orthologues for *M. pyrifera*, zero or one orthologue for *S. japonica*, and any number of orthologues for *S. polyschides* and *M. claveaformis*, identified 9,317 orthogroups (*Supplementary file 3*). The assumption underlying this approach was that the multiple gene models in the poor-quality genomes corresponded to fragments of a single gene, but note that we cannot rule out the possibility that a

minority of these groups of genes corresponded to tandem gene duplications. To calculate generation bias when a species possessed more than one gene model in a particular orthogroup, each model was given a score of +1, −1, or 0 if the expression data indicated that they were sporophyte-biased/specific, gametophyte-biased/specific or neither, respectively. The mean of these values was then calculated and the orthologue was classed as sporophyte-biased, gametophyte-biased or unbiased if the value was >0, <0, or 0, respectively.

## Algal strains and culture conditions

Wild type *Ectocarpus* species 7 strain Ec32 (Roscoff Culture Collection RCC4962) male gametophytes were grown in sterile Provasoli-enriched (0.5 x) natural sea water at 13 °C under 12 hr:12 hr day:night conditions until they reached maturity. Synchronous gamete release was induced by removing excess seawater and incubating in the dark and in the cold for 1 hr. Sterile Provasoli-enriched (0.5 x) natural sea water (*Starr and Zeikus, 1993*; *Coelho et al., 2012b*) was then added and the gametophytes incubated for 30 min to allow gamete release. The medium was filtered through 1 μm cell strainer (pluriSelect, Leipzig, Germany) to eliminate any contaminating gametophytic fragments. The gamete suspension was then diluted to a concentration of 1.5 million cells per mL ($A_{450nm} = 0.01$) to produce a density of 300 cells per mm² on plastic 150 cm Petri dishes after gamete settlement. After gamete release, the Petri dishes were incubated at 13 °C under 12 hr:12 hr day:night conditions at a light intensity of 40 μmol.m$^{-2}$.s$^{-1}$ for 24 or 48 hr. Cultures were sampled in the late afternoon to ensure homogeneity with regard to circadian rhythms. Sampling was carried out by scraping the bottom of the culture plate with a cell scraper (Sarstedt, Nümbrecht, Germany) after the seawater medium had been removed. The cell suspension was centrifuged at 14,000 × *g* for 15 min and the pellet flash-frozen in liquid nitrogen. All replicates were biological replicates.

## RNA extraction and RNA-seq sequencing and mapping

RNA was extracted using the Macherey Nagel Nucleospin Plant and Fungi kit (Hoerdt, France). Frozen pellets were resuspended in lysis buffer (500 μL of PFL buffer, 40 μL of PFR buffer and 50 μL of 1 M DTT per sample) and the cells were lysed by three 30 s treatments at 7800 rpm in 2 mL 'tough microorganism' lysis tubes (VK05) in a Precellys Evolution homogeniser (Bertin Technologies, Montigny-le-Bretonneux, France), followed by a 5 min incubation at 56 °C. RNA extraction followed the manufacturer's instructions except that a DNA digestion step was added after the first wash (digestion on the column with 5 μL of Qiagen DNAse I in 75 μL RDD buffer for 20 s before a second wash with PFW I buffer). The PFW II buffer wash step was carried out three times. RNA was eluted in 50 μL of RNAse-free $H_2O$. RNA-seq libraries were constructed after oligo-dT selection and sequenced by Genewiz (Leipzig, Germany) on an Illumina NovaSeq 6000, using a standard unstranded paired-end protocol to produce a minimum of 50 million reads per condition. Data quality was assessed with FastQC (*Andrews, 2016*) version 0.11.0 or 0.11.9 and reads were trimmed with Trim Galore (*Krueger, 2015*) version 0.6.5 with parameters --illumina --length 50 --quality 24 --stringency 6 --max_n 3. Genomes were indexed and data was mapped onto the v2 version of the *Ectocarpus* species 7 reference genome (*Sterck et al., 2012*; *Cormier et al., 2017*) using HISAT2 (*Kim et al., 2019*) version 2.1.0 with default parameters (the parameter --rna-strandedness was specified according to the information in *Supplementary file 1*). The output was converted to bam format and ordered and indexed using the SAMtools (*Danecek et al., 2021*) version 1.10 functions order and index. Quantification of transcript abundance was carried out using the featureCounts function (*Liao et al., 2014*) from the Subread package version 2.0.1 on CDS features grouped by Parent or LocusID.

## Identification of differentially expressed genes

Differentially expressed genes were identified using DESeq2 (*Love et al., 2014*) version 1.34.0 in R version 4.1.2. Genes were considered to be differentially expressed if they had an adjusted (Benjamini-Hochberg false discovery rate) p-value of <0.05 and an absolute log$_2$(FoldChange) greater than 1. Breakpoints in gene expression trends were analysed using the Trendy R package (*Bacher et al., 2018*) version 1.16.0 with the functions trendy (maxK = 3, pvalCut = 0.05) and topTrendy (adjR2Cut = 0.5). DESeq2 version 1.34.0 was also used to calculate normalised counts (NormalisedCounts) using the median of ratios method. Transcripts per kilobase million (TPM) values were calculated from counts of mapped reads using an in-house custom script (*Gaschet, 2019*). Based on the differential

expression analyses and the TPM values, genes were classified into six categories: gametophyte-biased (mean gametophyte and mean sporophyte TPM values both ⩾1, log2(fold change) ⩾1, adjusted p-value<0.05), sporophyte-biased (mean gametophyte and mean sporophyte TPM values both ⩾1, log2(fold change) ⩽−1, adjusted p-value<0.05), gametophyte-specific (mean TPM <1 for the sporophyte and ⩾1 for the gametophyte, log2(fold change) ⩾1, adjusted p-value<0.05), sporophyte-specific (mean TPM <1 for the gametophyte and ⩾1 for the sporophyte, log2(fold change) ⩽−1, adjusted p-value<0.05), unbiased genes (mean gametophyte and mean sporophyte TPM values ⩾ 1, log2(fold change)<1 or >-1 and/or adjusted p-value ⩾ 0.05) and unexpressed genes (mean gametophyte and mean sporophyte TPM values both <1).

## Gene ontology and prediction of subcellular localisation

Gene ontology term annotations were extracted from the Interproscan (*Jones et al., 2014*) annotation files generated by the Phaeoexplorer consortium (https://phaeoexplorer.sb-roscoff.fr/downloads/; *Denoeud et al., 2024*). The subcellular localisation of proteins was predicted by identifying amino-terminal targeting peptides using HECTAR version 1.3 (*Gschloessl et al., 2008*).

## Calculation of breadth of expression and Jaccard indices

Breadth of expression (tau) was calculated in R based on the following formula:

$$\tau = \left( \sum (1 - \hat{x}_i) \right) / (n - 1)$$

where $\hat{x}_i$ is the expression level of gene $i$ normalized by the maximum expression value across all conditions (mean log$_2$(NormalisedCounts +1)), and $n$ is the number of conditions.

Jaccard indices were calculated and plotted using vegan version 2.6.8 (https://cran.r-project.org/web/packages/vegan/index.html).

## Enrichment analysis

Enrichment of co-expression modules in differentially-expressed genes, predicted subcellular localisations or gene ontology terms was computed using the enricher function from the R package clusterProfiler (*Yu et al., 2012*) version 4.2.2. p-values were adjusted for multiple testing based on the Benjamini-Hochberg false discovery rate.

## Identification of modules of co-expressed genes

For both the *Ectocarpus* species 7 and the *D. dichotoma* datasets, raw counts were normalised using the estimateSizeFactors and counts (with option normalized = TRUE) functions from the R package DESeq2 (*Love et al., 2014*) version 1.34.0. Genes with a normalised count below 45 in all the samples analysed for the relevant species were removed. The normalised counts matrix was then transformed with the log1p (=log$_2$(x+1)) R function. To fit a scale-free topology model, the soft-thresholding power was selected using the pickSoftThreshold function, and gene modules were built using the blockwise-Modules function from the WGCNA R package (*Langfelder and Horvath, 2008*) version 1.71. For *Ectocarpus* species 7, this function was run with options power = 16, maxBlockSize = 16,100 (which yields a topological matrix in one block), minModuleSize = 30, reassignThreshold = 0, mergeCutHeight = 0.2, pamRespectsDendro = FALSE, randomSeed = T, corType = "Pearson". For *D. dichotoma*, the function was run with power = 18, maxBlockSize = 16,000 (which yields a topological matrix in one block), minModuleSize = 30, reassignThreshold = 0, mergeCutHeight = 0.25, pamRespectsDendro = FALSE, randomSeed = T, corType = 'Pearson'. Complete module annotation and module membership (MM) scores (or kME, representing the correlation between the gene and the module eigengene) can be found, together with associated p-values, in *Supplementary file 6* and *Supplementary file 8* D. dichotoma gene module descriptions are in *Figure 6—figure supplement 1B*.

## Conservation of gene co-expression modules between *Ectocarpus* species 7 and *D. dichotoma*

An orthologue analysis was carried out specifically for *Ectocarpus* species 7 and *D. dichotoma* using Orthofinder (*Emms and Kelly, 2019*) version 2.5.2 to optimise the identification of one-to-one orthologues for these two species (*Supplementary file 7*). The one-to-one orthologue set generated by this

analysis was then used to evaluate module conservation across the two species. The WGCNA package modulePreservation function (networkType = 'signed', nPermutations = 20, maxGoldModuleSize = 1000, maxModuleSize = 2500) and the NetRep package (*Ritchie et al., 2016*) version 1.2.7 moduleP-reservation function (nPerm = 10,000, on the WGCNA-computed correlation and TOM matrices) were used to compute module preservation statistics. In addition, the silhouette coefficient for modules was computed using the silhouette_SimilarityMatrix function from the CancerSubtypes R package (*Xu et al., 2017*) version 1.20.0 on the correlation matrix. All values can be found in *Supplementary file 8*. The gene correlation heatmap (*Figure 6B*) was produced using the NetRep (*Ritchie et al., 2016*) version 1.2.7 plotCorrelation function. The gold module was a random sample of 1000 *Ectocarpus* species 7 genes that provided a control distribution of gene expression patterns for comparisons with *D. dichotoma* modules. Cross-species module comparisons were visualised as a contingency table (*Figure 6C*) constructed by interpolating orthologous gene counts between the *Ectocarpus* species 7 and *D. dichotoma* datasets. For each pair of modules, a list of orthologues present in at least one of the two modules was established and a Fisher's exact test was performed on Boolean vectors indicating whether each gene was present in each of the two modules to assess the statistical association between module assignment in both species. The objective was therefore to determine whether each pair of *Ectocarpus* species 7 and *D. dichotoma* modules shared a greater number of one-to-one orthologues than expected from a random distribution. Fisher's exact test was applied using the fisher.test function from the R stats package (version 4.1.2). Cells in the contingency table were colour-coded according to $-\log_{10}$(p-value).

The contingency table (*Figure 6C*) was constructed by interpolating gene counts between the *Ectocarpus* species 7 and *D. dichotoma* datasets and using the fisher.test function (alternative = 'greater') from the R stats package (version 4.1.2).

### Detection of conservation of gene expression patterns between *Ectocarpus* species 7 and *D. dichotoma*

To assess gene expression pattern preservation between *Ectocarpus* species 7 and *D. dichotoma*, a z-score (mean for one condition – mean across all conditions / standard deviation) was computed from the $\log_2$(NormalisedCounts +1) values for each of the six common timepoints between the *Ectocarpus* species 7 and *D. dichotoma* datasets (*Supplementary file 1*). A Pearson correlation coefficient was then calculated using the R stats package (version 4.1.2) cor.test function (alternative = 'greater', method = 'pearson').

### Manual reannotation of ribosomal protein genes and three-dimensional representation of ribosomal proteins

*Ectocarpus* species 7 ribosomal proteins were manually annotated using the recent *Arabidopsis thaliana* nomenclature (*Scarpin et al., 2023*) as a reference (*Supplementary file 12*). *Ectocarpus* species 7 ribosomal proteins were compared with the *A. thaliana* ribosomal protein dataset using BLASTp and the annotations transferred based on the best BLASTp match. To visualise differentially expressed genes in relation to the structure of the ribosome, *Ectocarpus* species 7 ribosomal proteins were mapped onto the *Triticum aestivum* ribosome structure (Protein data bank model 4V7E; *Gogala et al., 2014*) and the mapping was visualised using UCSF ChimeraX version 1.9 (*Meng et al., 2023*).

### General data treatment and figure preparation tools

Several R modules were used at multiple stages of the study. These included rtracklayer (*Lawrence et al., 2009*) version 1.52.1, Tidyverse (*Wickham et al., 2019*) version 2.0.0, and sjmisc (*Lüdecke, 2018*) version 2.8.9, which were used to handle and analyse data, and ggplot2 (*Wickham et al., 2016*) version 3.4.2 and Pheatmap (*Kolde, 2019*) version 1.0.12, which were used to generate figures.

## Acknowledgements

We thank Julia Morales for helpful discussions about ribosomal proteins, Laurence Dartevelle and Elodie Rolland for help with algal cultures, Delphine Scornet, Masakazu Hoshino, Inka Bartsch, Akira F Peters, Samuel Boscq, Shannon DeVanney for providing seaweed photographs. We are grateful to the Roscoff Bioinformatics platform ABiMS (http://abims.sb-roscoff.fr), which is part of the Institut

Français de Bioinformatique (ANR-11-INBS-0013) and BioGenouest network, for providing both help and computing and storage resources. This work was supported by the ANR project Epicycle (ANR-19-CE20-0028-01), the France Génomique National infrastructure project Phaeoexplorer (ANR-10-INBS-09), the CNRS and Sorbonne University. PR acknowledges support from the PhD funding program of École Normale Supérieure de Lyon.

## Additional information

### Funding

| Funder | Grant reference number | Author |
| --- | --- | --- |
| Agence Nationale de la Recherche | ANR-19-CE20-0028-01 | J Mark Cock |
| Agence Nationale de la Recherche | ANR-10-INBS-09 | J Mark Cock |
| École Normale Supérieure de Lyon | PhD funding program | Pélagie Ratchinski |

The funders had no role in study design, data collection and interpretation, or the decision to submit the work for publication.

### Author contributions

Pélagie Ratchinski, Conceptualization, Data curation, Formal analysis, Investigation, Visualization, Writing – original draft, Writing – review and editing; Olivier Godfroy, Conceptualization, Data curation, Formal analysis, Supervision, Investigation, Visualization, Writing – review and editing; Benjamin Noel, Jean-Marc Aury, Resources, Writing – review and editing; J Mark Cock, Conceptualization, Supervision, Funding acquisition, Project administration, Writing – review and editing

### Author ORCIDs

Pélagie Ratchinski ⓘ https://orcid.org/0009-0009-0609-2675
Jean-Marc Aury ⓘ https://orcid.org/0000-0003-1718-3010
J Mark Cock ⓘ https://orcid.org/0000-0002-2650-0383

Reviewer #2 (Public review): https://doi.org/10.7554/eLife.107449.3.sa1
Author response https://doi.org/10.7554/eLife.107449.3.sa2

## Additional files

### Supplementary files

Supplementary file 1. RNA-seq data used in this study. GA, gametophyte; SP, sporophyte; pSP, partheno-sporophyte; GBG, generation-biased genes.

Supplementary file 2. Generation-biased and generation-specific gene sets in 10 species of brown algae based on comparisons of adult sporophyte and gametophyte generations. (A) Overview of the generation-biased and generation-specific gene sets of each species. (B) Complete proteomes of the 10 species indicating the results of the comparison of adult sporophytes and gametophytes, including $\log_2$(fold change), $p$-value, mean TPM and assigned generation-biased expression class. TPM, transcripts per kilobase million.

Supplementary file 3. List of orthogroups for the 10 brown algal species analysed for generation-biased gene expression. The genomes analysed (with ENA accession numbers or links to ORCAE in brackets) correspond to: *Dictyota dichotoma* strain ODC1387m (GCA_964200555), *Ectocarpus* species 7 strain Ec32 (https://bioinformatics.psb.ugent.be/orcae/overview/EctsiV2), *Macrocystis pyrifera* strain P11B4 (GCA_964200385), *Myriotrichia clavaeformis* strain Myr cla04 (GCA_964200105), *Pylaiella littoralis* strain U1.48 (GCA_964200295), *Saccharina latissima* strain SLPER63f7 (GCA_964200175), *Saccharina japonica* strain Ja (https://bioinformatics.psb.ugent.be/orcae/overview/Sacja), *Saccorhiza polyschides* strain SpolBR94m (GCA_964200605), *Scytosiphon*

*promiscuus* strain Ot110409-Otamoi-16-male (GCA_964200365) and *Sphacelaria rigidula* strain Sph rig Cal Mo 4-1-68b (GCA_964200075). The one-to-one orthologues correspond to the 9,317 orthogroups identified using the relaxed criteria described in the Materials and methods.

Supplementary file 4. Counts of RNA-seq reads mapped to *Ectocarpus* species 7 genes. This data was used to calculate the gene expression levels reported in *Supplementary file 5*.

Supplementary file 5. Functional and expression-related information for *Ectocarpus* species 7 genes. List of all *Ectocarpus* species 7 genes with transcript abundance under each of the conditions analysed, measured as $\log_2$(NormalisedCounts +1), WGCNA module assignment (Colour), WGCNA module membership (MM) scores and associated *p*-values (pMM) for each gene in each module, the functional annotation for each gene (description), the manually assigned functional category (Manual.functional.category), the DESeq2 output for the comparison between free-swimming male gamete and sporophyte initial cell stage (gamete_initial), and adult sporophyte and gametophyte (sporophyte_gametophyte) with the corresponding differential expression (DE) annotation, the HECTAR output for each gene with the predicted targeting category and corresponding scores.

Supplementary file 6. Overview of the gene co-expression modules for *Ectocarpus* species 7 and *D. dichotoma*. The grey modules contain all the genes that could not be assigned to a co-expression module. The gold module was a random sample of 1000 *Ectocarpus* species 7 genes that provided a null distribution of gene expression patterns for the comparisons with *D. dichotoma* modules.

Supplementary file 7. Orthofinder orthogroup analysis of *Ectocarpus* species 7 and *D. dichotoma*. The genomes analysed (with ENA accession numbers or links to ORCAE in brackets) correspond to: *Dictyota dichotoma* strain ODC1387m (GCA_964200555) and *Ectocarpus* species 7 strain Ec32 (https://bioinformatics.psb.ugent.be/orcae/overview/EctsiV2).

Supplementary file 8. Conservation of life-cycle-related co-expression modules between *Ectocarpus* species 7 and *D. dichotoma*. For each *Ectocarpus* species 7 module: preservation statistics computed by the WGCNA modulePreservation function (200 permutations), preservation statistics computed by the NetRep modulePreservation function (10,000 permutations) with the associated $\log_{10}$(p-value) (Bonferroni-corrected for WGCNA), and the silhouette score for each module in *D. dichotoma*. WGCNA medianRank statistic, aggregation of the densityMedianRank and connectivityMedianRank statistics; WGCNA – medianRankDensity.pres, WGCNA density median rank preservation statistic; WGCNA – medianRankConnectivity.pres, WGCNA connectivity median rank preservation statistic; WGCNA propVarExplained.pres, coherence: proportion of module variance explained by the module eigengene (=summaryprofile = eigenvector of the 1st principal component across all observations for every node composing the module); WGCNA – meanSignAwareKME.pres, average node contribution: average Pearson correlation coefficient to the module's summary profile; WGCNA - meanSignAwareCorDat.pres, density of correlation structure: how strongly modules are correlated in the test dataset, average node correlation; WGCNA - meanAdj.pres, average edge weight: average connection strength between nodes; WGCNA – meanMAR.pres, average module adjacency ratio; WGCNA: cor.kIM, concordance of intramodular connectivity: similarity of the relative rank of each nodes' weighted degree (intramodular connectivity) across datasets; WGCNA – cor.kME, concordance of node contribution: preservation of relative rank of nodes (ordered by Pearson correlation coefficient to the module's summary profile) across datasets; WGCNA – cor.kMEall; concordance of node contribution: preservation of relative rank of nodes (ordered by Pearson correlation coefficient to the module's summary profile) across datasets taking into account the kME of the gene for all modules; WGCNA - cor.cor, concordance of correlation structure: quantifies how similar the correlation structure is across datasets; WGCNA – cor.MAR, concordance of module adjacency ratio: concordance of the ratio between the maximum and the minimum adjacency in the module; WGCNA - separability.pres, separability preservation; NetRep - avg.weight, average edge weight: average connection strength between nodes; NetRep - coherence, coherence, i.e. proportion of module variance explained by the module eigengene (=summaryprofile = eigenvector of the 1st principal component across all observations for every node composing the module); NetRep – cor.cor, concordance of correlation structure: quantifies how similar the correlation structure is across datasets; NetRep - cor.degree, concordance of intramodular connectivity: similarity of the relative rank of each nodes' weighted degree (intramodular connectivity) across datasets; NetRep - cor.contrib, concordance of node contribution: preservation of relative rank of nodes (ordered by Pearson correlation coefficient to the module's summary profile) across datasets; NetRep - avg.cor, density of correlation structure: how strongly modules are correlated in the test dataset, average node correlation; NetRep - avg.contrib, average node contribution: average Pearson correlation coefficient to the module's summary profile; silhouette.cluster.width, Silhouette coefficient; silhouette.distance.to.average, Silhouette coefficient.

The 'gold' module was a random sample of 1000 *Ectocarpus* species 7 genes that provided a control distribution of gene expression patterns for comparisons with *D. dichotoma* modules. The 'dark turquoise' module was not included in this analysis because it contained zero orthologues (*Supplementary file 6*).

Supplementary file 9. Counts of RNA-seq reads mapped to *D. dichotoma* genes. This data was used to calculate the gene expression levels reported in *Supplementary file 10*.

Supplementary file 10. Functional and expression-related information for *D. dichotoma* genes. List of all *D. dichotoma* genes with the $\log_2$(NormalisedCounts +1) under each condition used for the analysis, WGCNA module assignment (Colour), WGCNA module membership (MM) scores and associated p-values (pMM) for each gene in each module.

Supplementary file 11. Correlation between the expression patterns of orthologous transcription factors in *Ectocarpus* species 7 and *D. dichotoma*. Correlation was evaluated based on comparisons of six approximately equivalent stages for the two species (*Supplementary file 1*). The names of the *Ectocarpus* species 7 and *D. dichotoma* genes in each orthogroup are given in *Figure 6—figure supplement 3*. padj, p-value adjusted for multiple testing based on the Benjamini-Hochberg false discovery rate.

Supplementary file 12. Manual re-annotation of the *Ectocarpus* species 7 ribosomal proteins. The e-values and scores are for BLASTp queries of the *Ectocarpus* species 7 ribosomal proteins against the *Arabidopsis thaliana* ribosomal protein dataset.

MDAR checklist

### Data availability

All the new sequence data generated by this study have been deposited in the European Bioinformatics Institute/European Nucleotide Archive (EBI/ENA) database and are publicly available (see *Supplementary file 1* for details).

The following dataset was generated:

| Author(s) | Year | Dataset title | Dataset URL | Database and Identifier |
| --- | --- | --- | --- | --- |
| Ratchinski P, Godfroy O, Noel B, Aury JM, Cock JM | 2025 | Transcriptomics of early partheno-sporophyte development of Ectocarpus sp7 | https://www.ebi.ac.uk/ena/browser/view/PRJEB79549 | EBI European Nucleotide Archive, PRJEB79549 |

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
