## [Editor Report · eLife Assessment]

This manuscript presents an in-depth analysis of gene expression across multiple brown algal species with differing life histories, providing **convincing** evidence for the conservation of life cycle-specific gene expression. While largely descriptive, the study is an **important** step forward in understanding the core cellular processes that differ between life cycle phases, and its findings will be of broad interest to developmental and evolutionary biologists.

---

## [Referee Report · Reviewer #2 (Public review)]

Summary:

The manuscript by Ratchinski et al presents a comprehensive analysis of developmental and life history gene expression patterns in brown algal species. The manuscript shows that the degree of generation bias or generation-specific gene expression correlates with the degree of dimorphism. It also reports conservation of life cycle features within generations and marked changes in gene expression patterns in Ectocarpus in the transition between gamete and early sporophyte. The manuscript also reports considerable conservation of gene expression modules between two representative species, particularly in genes associated with conserved functional characteristics.

Strengths:

The manuscript represents a considerable "tour de force" dataset and analytical effort. While the data presented is largely descriptive, it is likely to provide a very useful resource for studies of brown algal development and for comparative studies with other developmental and life cycle systems.

Comments on revisions

The authors have provided in their response (point 1) a good clarification for their rationale in excluding fucoid algae from the study, based on the diploid nature of the fucoid life cycle. Similarly, they have noted (point 2) that "the relationship between changes in gene expression during very early sporophyte development and during alternation of life cycle generations could be investigated further using a highlydimorphic kelp model system such as *Saccharina latissima*." For the benefit of the reader who may not be too familiar with the different life cycles in brown algae, I would recommend that these clarifications are included in the Discussion.

Otherwise the authors have addressed my previous comments adequately.

---

## [Author Response]

The following is the authors’ response to the original reviews.

**Reviewer #1 (Public review):**
Summary:The authors have examined gene expression between life cycle stages in a range of brown macroalgae to examine whether there are conserved aspects of biological features.Strengths:The manuscript incorporates large gene expression datasets from 10 different species and therefore enables a comprehensive assessment of the degree of conservation of different aspects of gene expression and underlying biology.The findings represent an important step forward in our understanding of the core aspects of cell biology that differ between life cycle phases and provide a substantial resource for further detailed studies in this area. Convincing evidence is provided for the conservation of lifecycle-specific gene expression between species, particularly in core housekeeping gene modules.Weaknesses:I found a few weaknesses in the methodology and experimental design. I think the manuscript could have been clearer when linking the findings to the biology of the brown algae.
**Reviewer #2 (Public review):**
Summary:The manuscript by Ratchinski et al presents a comprehensive analysis of developmental and life history gene expression patterns in brown algal species. The manuscript shows that the degree of generation bias or generation-specific gene expression correlates with the degree of dimorphism. It also reports conservation of life cycle features within generations and marked changes in gene expression patterns in Ectocarpus in the transition between gamete and early sporophyte. The manuscript also reports considerable conservation of gene expression modules between two representative species, particularly in genes associated with conserved functional characteristics.Strengths:The manuscript represents a considerable "tour de force" dataset and analytical effort. While the data presented is largely descriptive, it is likely to provide a very useful resource for studies of brown algal development and for comparative studies with other developmental and life cycle systems.Weaknesses:Notwithstanding the well-known issues associated with inferring function from transcriptomics-only studies, no major weaknesses were identified by this reviewer.
**Reviewing Editor Comments:**
The overall assessment of the reviewers does not contain major aspects of concern. We nevertheless recommend that the authors carefully consider the constructive comments, as this will further improve their manuscript.
**Reviewer #1 (Recommendations for the authors):**
(1) Line 32: The abstract states 'considerable conservation of co-expressed gene modules', but the degree of conservation between Ectocarpus and D. dichotoma appeared limited to specific subsets of genes with highly conserved housekeeping functions, e.g., translation. I think the wording of the abstract should be rephrased to better reflect this.

We agree that genes with housekeeping functions figure strongly in the gene modules that showed strong conservation between *Ectocarpus* species 7 and *D. dichotoma* (and we actually highlight this point in the manuscript) but we do not believe that this invalidates the conservation. In the analysis shown in Figure 6A, for example, high scores were obtained for both connectivity and density for about a third of the gene modules and these modules cover broad range of cellular functions. This is a significant result given the large phylogenetic distance and we feel that "considerable conservation" is appropriate as a description of the level of correlation.

(2) Introduction - The Introduction needs a better explanation of the biology of the life cycle phases. Some of this information is present in the 1st paragraph of Materials and Methods, although it would be preferable to include this information within the main text, ideally within the Introduction before the Results are described. For example, when are flagella present? The presence of flagella could be indicated in Figure 3. The ecology of the life cycle is also not described. Are life cycles present in the same ecological niche? Do they co-exist or occupy distinct environments? It would be useful to understand how the observed genotypes could relate to this wider aspect of the brown algal biology.

We have added a sentence to explain that zoids (gametes and spores) are the only flagellated stages of the life cycle (line 678). In addition, in the legend for Figure 3, we have indicated which of the life cycle stages analysed in panel 3A consisted entirely or partially of flagellated cells. We have also added information about phenology to the Introduction.

(3) Line 127. 'The proportion of generation specific genes was positively correlated with the level of dimorphism'. The level of dimorphism between species was not clear to me. This needs to be clearly displayed in Figure 1B.

We had attempted to illustrate the level of dimorphism, using the size of each generation as a measurable proxy, in Figure S1 but we agree that the information was not very clearly presented. To improve clarity, we now provide independent size scales for each generation of the life cycle in this figure and state in the legend that "Size bars indicate the approximate sizes of each generation of each life cycle, providing an indication of the degree of dimorphism between the two generations.". In the text, Figure S1 is cited earlier in the paragraph but we now repeat the citation of the figure at the end of the sentence "The proportion of generation-specific genes (...) was positively correlated with the level of dimorphism" so that the reader can specifically consult the supplementary figure for this phenotypic parameter.

(4) Line 267. Are there known differences in cell wall composition between life cycle phases or within each generation as individual life cycle phases mature (e.g., differences between unicellular and multicellular stages)?

Detailed comparative analyses of cell wall composition at different stages of the life cycle have not been carried out for brown algae. However, Congo red stains *Ectocarpus* gametophytes but not sporophytes (Coelho *et al*., 2011), indicating a difference in cell wall composition between the two generations. Zoids (spores and gametes) do not have a cell wall and calcofluor white staining of meio-spores has indicated that a cell wall only starts to be deposited 24-48 hours post-release (Arun *et al*., 2013).

(5) Line 388. The authors should comment on the accuracy of OrthoFinder for different gene types across this degree of divergence (250 MYA). The best conservation was found in genes with housekeeping characteristics (line 401). It may be that these gene modules show the highest degree of conservation in expression patterns, but I also wonder whether they pattern may also emerge because finding true orthologues is easier for highly conserved gene families.

We do not believe that this is the case because, as mentioned above, the "housekeeping" modules cover quite a broad range of cellular functions. Note also that the modules were given functional labels based on their being clearly enriched in genes corresponding to a particular class of function but not all the genes in a module have a predicted function that corresponds to the functional classification.

However, we have carried out an analysis to look for evidence of the bias proposed by the reviewer. For this, we used BLASTp identity scores as an approximate proxy for pairwise identity between *Ectocarpus* species 7 and *D. dichotoma* one-to-one orthologues in each module and plotted the mean identity score for each module against the Fischer test p-value of the contingency table in Figure 6C (Author response image 1).

**Author response image 1. sa2fig1:** Plot of estimations of the mean percent shared identity between the orthologues within each module (based on mean BLASTp identity scores) against log10(pvalue) values obtained with the Fisher's exact test applied in Figure 6C to determine whether pairs of modules shared a greater number of one-to-one orthologues than expected from a random distribution. Error bars indicate the standard deviation.

This analysis did not detect any correlation between the degree of sequence conservation of orthologues in a module and the degree of conservation of the module between *Ectocarpus* species 7 and *D. dichotoma*.

Minor comments(1) Line 650 loose should be lose.

The error has been corrected.

(2) Line 695 filtered through a 1 μm filter to remove multicellular gametophyte fractions. Is this correct? It seems too small to allow gametes to pass through.

Yes, the text is correct, a 1 μm filter was used. The gametes do pass through this filter, presumably because they do not have a rigid cell wall, allowing them to squeeze through the filter when a light pressure is applied.

(3) Line 709 - DDT should be DTT

The error has been corrected.

**Reviewer #2 (Recommendations for the authors):**
(1) It is not clear why the chosen species for analysis do not include fucoid algae, which display a high degree of dimorphism between generations and which are relatively well studied with respect to gene expression patterns during early development. Indeed, it was recently shown that gene expression patterns in developing embryos of Fucus spp. obey the "hourglass" pattern whereby gene expression shows a minima of transcription age index (i.e., higher expression of evolutionarily older genes) associated with differentiation at the phylotypic stage. I am somewhat surprised that the manuscript does not consider this feature in the analysis or discussion.

Brown algae of the order Fucales have diploid life cycles and therefore do not alternate between a sporophyte and gametophyte generation. It is for this reason that we thought that it was more interesting to compare *Ectocarpus* species 7 with *D. dichotoma*, which has a haploid-diploid life cycle.

(2) In Discussion, the comparison of maternal to zygote transition in animals and land plants, which show a high degree of dimorphism, with Ectocarpus would be strengthened by data/discussion from other brown algae that show a high degree of dimorphism.

Animals have diploid life cycles and dimorphism in that lineage generally refers to sexual rather than generational dimorphism. Land plants do have highly dimorphic haploiddiploid life cycles but it is unclear how this characteristic relates to events that occur during the maternal to zygote transition. In Ectocarpus, the transition from gamete to the first stages of sporophyte development involved more marked changes in gene expression than we observed when comparing the mature sporophyte and gametophyte generations (Figure 3C). At present, there is no evidence that events during these two transitions are correlated. The relationship between changes in gene expression during very early sporophyte development and during alternation of life cycle generations could be investigated further using a highly dimorphic kelp model system such as *Saccharina latissima* but we are not aware of any studies that have specifically addressed this point.

(3) Since marked changes were observed during the transition from gamete to early sporophyte in Ectocarpus, it would be interesting to know how gene expression patterns change during the transition from gamete to partheno-sporophyte. Would the same patterns of downregulation and upregulation be expected?

The sporophyte individuals derived from gamete parthenogenesis (parthenosporophytes) are indistinguishable morphologically and functionally from diploid sporophytes derived from gamete fusions (see line 76). They also express generation marker genes in a comparable manner (Peters *et al*., 2008). Based on these observations, we have treated partheno-sporophytes and diploid sporophytes as equivalent in our experiments. For clarity, we have now distinguished partheno-sporophyte from diploid sporophyte samples in Table S1.

(4) The authors show a correlation between the degree of dimorphism and generation-biased or generation-specific expression. How was the degree of dimorphism quantified?

The degree of dimorphism is illustrated in Figure S1 using the relative size of the two generations as a proxy. Size estimations are approximate because the size of an individual of a particular species is quite variable but the ten species nonetheless represent a very clear gradient of dimorphism due to the extreme differences in size between generations of species at each end of the scale, with the sporophyte generation being several orders of magnitude larger than the gametophyte generation or visa versa.

References

Arun A, Peters NT, Scornet D, Peters AF, Cock JM, Coelho SM. 2013. Non-cell autonomous regulation of life cycle transitions in the model brown alga *Ectocarpus*. New Phytol 197:503– 510. doi:10.1111/nph.12007

Coelho SM, Godfroy O, Arun A, Le Corguillé G, Peters AF, Cock JM. 2011. OUROBOROS is a master regulator of the gametophyte to sporophyte life cycle transition in the brown alga *Ectocarpus*. Proc Natl Acad Sci USA 108:11518–11523. doi:10.1073/pnas.1102274108

Peters AF, Scornet D, Ratin M, Charrier B, Monnier A, Merrien Y, Corre E, Coelho SM, Cock JM. 2008. Life-cycle-generation-specific developmental processes are modified in the *immediate upright* mutant of the brown alga *Ectocarpus siliculosus*. Development 135:1503–1512.doi:10.1242/dev.016303